# Explaining Node Embeddings

**Zohair Shafi**                                                                    *shafi.z@northeastern.edu*
*Northeastern University*
*Boston, MA, USA*

**Ayan Chatterjee**                                                              *chatterjee.ay@northeastern.edu*
*Northeastern University*
*Boston, MA, USA*

**Tina Eliassi-Rad**                                                              *t.eliassirad@northeastern.edu*
*Northeastern University*
*Boston, MA, USA*

**Reviewed on OpenReview:** *https://openreview.net/forum?id=QQZ8uPxFb3*

## Abstract

Node embedding algorithms produce low-dimensional latent representations of nodes in a graph. These embeddings are often used for downstream tasks, such as node classification and link prediction. In this paper, we investigate the following two questions: (Q1) Can we explain each embedding dimension with human-understandable graph features (e.g. degree, clustering coefficient and PageRank). (Q2) How can we modify existing node embedding algorithms to produce embeddings that can be easily explained by human-understandable graph features? We find that the answer to Q1 is yes and introduce a new framework called XM (short for eXplain eMbedding) to answer Q2. A key aspect of XM involves minimizing the nuclear norm of the generated explanations. We show that by minimizing the nuclear norm, we minimize the lower bound on the entropy of the generated explanations. We test XM on a variety of real-world graphs and show that XM not only preserves the performance of existing node embedding methods, but also enhances their explainability.

## 1 Introduction

Graph datasets are ubiquitous from social, to physical, and biological networks. The abundance of graph data has inspired the development of models and algorithms in the field of graph machine learning (Chami et al., 2022), a popular task being node embedding,[1] where nodes are embedded in a low-dimensional latent space. Many algorithms exist for embedding a graph's nodes into a low-dimensional space. Zhang et al. (2021a) compared a variety of such algorithms on tasks such as greedy routing and link prediction. Recently, much attention has been devoted to explainable graph machine learning (Burkart & Huber, 2021), in part because the performance of such models can often be misleading (Stolman et al., 2022), (Menand & Seshadhri, 2024). Recent work on explaining graph machine learning revolves around explaining decisions made on downstream tasks (Pope et al., 2019), (Baldassarre & Azizpour, 2019), (Ying et al., 2019), (Luo et al., 2020), (Vu & Thai, 2020), Yuan et al. (2021), (Liu et al., 2018)]. An outlier here is Gogoglou et al. (2019)'s work. They investigate individual dimensions of an embedding and define an interpretability score for each dimension to measure how well each dimension defines a subgroup of nodes (see Section 2 for details).

In this work, our aim is to bridge the gap between human-understandable features and node embeddings in graphs. In natural language processing (NLP), each dimension of an embedding can often be associated with a well-defined, interpretable concept. For example, a dimension might capture the concept of gender, as demonstrated by the famous "king and queen" analogy (Ethayarajh et al., 2018). However, transferring

---

[1]Sometimes node embedding is referred to as graph embedding.

this idea to graph data is not straightforward. For example, consider a network of academic collaborations, such as the PubMed citation network (Roberts, 2001). If we embed this network into three dimensions using a traditional method, the first dimension might group nodes with many connections (high node degree), the second might cluster nodes by shared research topics (communities), and the third could isolate interdisciplinary nodes (high node centrality). However, interpreting these dimensions in terms of concrete and human-understandable features is not trivial, as each dimension typically encodes a complex mixture of several attributes. This lack of interpretability presents a significant obstacle, especially when understanding and trusting the model's output is just as important as prediction.

To address this, we use a glossary for graphs, comprising of well-established graph features such as node degree, clustering coefficient, node betweenness centrality, and other topological features. These features are the cornerstone of characterizing any graph dataset (Albert & Barabási, 2002; Newman, 2003) and have previously been used in graph machine learning tasks (Henderson et al., 2012; Khoshraftar et al., 2021; Rossi et al., 2018). These features serve as building blocks that can help explain the dimensions of graph embeddings. For example, a person node with a high degree is a person with many friends. We introduce a framework called XM (short for EXPLAIN EMBEDDING) that makes node embeddings explainable by associating each dimension of the embedding with these well-defined, human-understandable graph features. Following Henderson et al. (2012), we refer to these features as "sense" features. For example, one embedding dimension might correspond to the node's degree, while another might relate to its clustering coefficient or centrality. These features can be tailored to specific domains.

**Problem Definition**  We are given the following:

- A graph $G = (V, E, X)$, where $V$ represents the set of vertices (nodes), $E$ represents the set of edges (links), and $X$ represents the node feature matrix (with $|V| = n$ nodes)

- A graph-embedding algorithm $y = g_\theta(G) \in \mathbb{R}^{n \times d}$ that maps each node $k \in V$ to a $d-$dimensional vector $y_k$ parameterized by $\theta$

- A list of sense features: $\forall k \in V, \vec{f_k} \in \mathbb{R}^{f \times 1}$

For each node $k \in V$, XM learns an Explain matrix $E_k \in \mathbb{R}^{d \times f}$. Each row of the Explain matrix corresponds to a dimension and each column corresponds to a sense feature. Table 1 summarizes the notation used throughout this work.

Concretely, we present our investigation of the following two questions. **Q1: Can we explain dimensions of a node embedding method with human-understandable graph features?** Examples of such features are degree, clustering coefficient, eccentricity, and PageRank. These features can be any set of features that a practitioner decides to use and are not limited to graph features. We will refer to them as "sense" features. **Q2: Can we modify existing node embedding algorithms to produce embeddings that are explained with sense features? If so, how?** We present a new framework called XM (short for EXPLAIN EMBEDDING) to answer this question. XM adds two constraints to an existing objective function for node embedding: (a) sparsity w.r.t. explaining an embedding dimension and (b) orthogonality between embedding dimensions. We provide an ablation study to analyze the impact of each constraint. XM outputs explainable node embeddings that are tested on a variety of real-world graphs, achieving downstream task performances comparable to the state of the art.

Our main contributions are as follows:

- We present a method for understanding what each dimension of a node embedding method means, with respect to a set of human-understandable sense features defined on nodes. This helps us to understand the placement of a node in the embedding space. These explanations are independent of any downstream task and are in the form of an *Explain* matrix, whose rows represent embedding dimensions and whose columns represent sense features.

| Symbol | Description |
|---|---|
| $G$ | Graph |
| $V$ | Set of vertices of graph $G$ |
| $E$ | Set of edges of graph $G$ |
| $X$ | Node feature matrix for graph $G$ |
| $d$ | Number of dimensions used to embed graph $G$ |
| $\vec{y}_i$ | $d-$dimensional embedding vector of graph $G$ for node $i$ |
| $f$ | Number of sense features |
| $\vec{f}_i$ | Sense feature vector for node $i$ |
| $E_i$ | Explain matrix for node $i$ |
| $H(A)$ | Von Neumann entropy of matrix $A$ |
| $D(A||B)$ | Bregman Divergence between matrices $A$ and $B$ |

Table 1: Notation Table

- To evaluate Explain matrices, we use their nuclear norms and show that minimizing the nuclear norm of an Explain matrix removes noise and reduces the lower bound of its entropy, which is a desirable property.

- We introduce the XM framework, which modifies the objective function of any existing node embedding method to produce embeddings that provide better quality explanations. We analyze the impact of each of XM's constraints (sparsity and orthogonality) through an ablation study and show that using both constraints leads to the largest reduction in nuclear norms. We demonstrate the effectiveness of XM by modifying a number of embedding algorithms on different real-world datasets and evaluating them on the downstream task of link prediction.

## 2 Related Work

Interest in explaining machine learning models has attracted much attention in recent years, in part because the performance of such models can often be misleading (Burkart & Huber, 2021). With the increasing application of machine learning to graph data, the development of explainable models on graphs is becoming a necessity. Figure 1 provides a taxonomy of various explanation methods on graph machine leaning models. We divide the existing literature into supervised and unsupervised methods.

**Supervised Methods** Most of the work on explainable models for graph machine learning falls into this category and focuses on explaining predictions for downstream tasks (such as link prediction or node classification). We further divide supervised methods into two categories: (1) gradient-based explanations and (2) subgraph-based explanations. Examples of **gradient-based explanations** include the works of Pope et al. (2019) and Baldassarre & Azizpour (2019). They extend the concept of explainable predictions from image data to network data by using gradient-based approaches to visualize and understand the predictions of Graph Convolutional Networks (GCNs). There are numerous methods under **subgraph-based explanations**. These methods focus on finding a subgraph that contributes to a particular classification. We divide these methods into three groups:

1. Direct subgraph extraction:
    - Ying et al. (2019) introduce *GNNExplainer*, which tries to find the minimal subgraph and the most relevant subset of node features that are key to the prediction of the GNN for a given instance. *GNNExplainer* maximizes the mutual information between the original prediction and the prediction of the extracted subgraph. Since the explanations of *GNNExplainer* are customized for each instance, it cannot provide explanations for a set of instances.
    - Luo et al. (2020) extend *GNNExplainer* by introducing *PGExplainer*, which uses a generative probabilistic model for graph data to learn the underlying structures from the observed

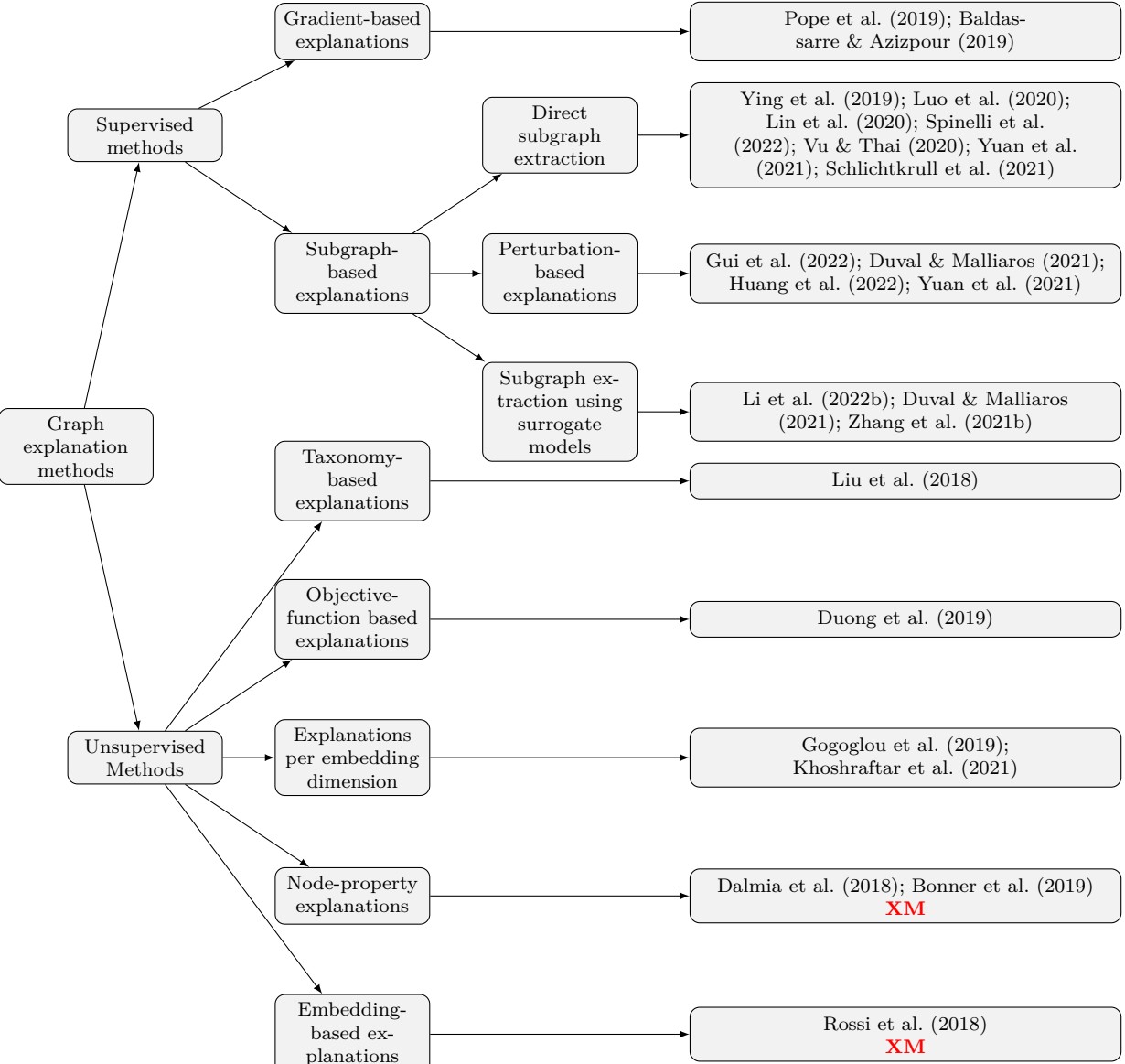

Figure 1: Taxonomy of various graph explanation methods. Section 2 provides a brief description for each method.

data. *PGExplainer* uses these underlying structures as explanations and models the underlying structure as edge distributions, sampling the explanatory graph. The generative model in *PGExplainer* is parameterized with a deep neural network to jointly explain the predictions of multiple instances.

- Spinelli et al. (2022) propose a meta-explainer to improve the quality of explanations during training time. Their goal is to steer the optimization procedure towards minima that allow post-hoc explainers such as *GNNExplainer* and *PGExplainer* to achieve better results.

- Vu & Thai (2020) introduce *PGM-Explainer*, which has similar goals as *GNNExplainer*: identify subgraphs and relevant features for the prediction of a particular instance. In contrast to *GNNExplainer*, *PGM-Explainer* uses probabilistic graphical models to explain substructures in a network. This allows *PGM-Explainer* to model uncertainty and provide probabilistic explanations.

- Yuan et al. (2021) use Monte Carlo tree search to search all possible subgraphs and identify substructures that are important to the downstream prediction task. To compute the importance of each subgraph to the prediction task, they use Shapley values (Lundberg & Lee, 2017).

- Schlichtkrull et al. (2021) learn a simple classifier that predicts if an edge can be dropped without deteriorating the model's performance. The remaining edges are then analyzed for interpretability.

2. Perturbation-based explanations: These methods primarily extend LIME (Ribeiro et al., 2016) and SHAP (Lundberg & Lee, 2017) to GNN.

- Gui et al. (2022) propose *FlowX*, which is a method for explaining the predictions of a GNN by identifying important message flows using Shapley values. A flow is defined as the set of $T$ messages that are passed around in a message-passing GNN with $T$ layers. Each flow for a given node is then treated as a player for the Shapley formulation.

- Huang et al. (2022) propose *GraphLIME*, which aims to explain the predictions of a GNN by (i) extending LIME (Ribeiro et al., 2016) to operate on graph data and (ii) using a surrogate model (e.g., a linear regression). *GraphLIME* provides local explanations because its perturbations are limited to the local neighborhood of nodes.

- Duval & Malliaros (2021) propose *GraphSVX*, whose goal is to provide explanations for the predictions of a GNN by using Shapley values and a surrogate model (e.g., a linear regression). Shapley values determine the "fair contributions" of each node, edge, or feature to the prediction of the model.

3. Subgraph extraction using surrogate models:

- Li et al. (2022b) use a student-teacher model to distill knowledge from a large pre-trained GNN to a shallow GNN, while explicitly learning the contribution weights between two nodes using an attention mechanism.

- Zhang et al. (2021b) propose *RelEx*, which identifies important nodes and links in the prediction of a given node. It treats the graph machine-learning model as a black box and learns relational explanations. Explanations come in the form of a mask matrix with sparsity constraints.

- Lin et al. (2020) also use sparsity constraints. They propose *GISST*, which combines an attention mechanism and sparsity regularization to yield the subgraph and node features that are key to the downstream task.

- Schnake et al. (2021) propose *GNN-LRP*, which uses layer-wise relevance propagation and outputs a collection of walks that are relevant for a given prediction.

A useful tool under supervised methods is *GraphXAI* (Agarwal et al., 2023). It provides a framework for GNN explainers. In particular, GraphXAI provides a synthetic graph generator called *ShapeGGen graph generator*. Their generator generates ground-truth explanations for homophilic, heterophilic, and attributed graphs. We refer the reader to a survey by Li et al. (2022a) on supervised methods for explaining graph machine learning models.

**Unsupervised Methods**  Our work is independent of the downstream task. Under unsupervised methods, we have five categories:

1. Taxonomy-based explanations: Liu et al. (2018) use network homophily and hierarchically cluster nodes based on the embeddings to create a taxonomy. Explanations come in the form of items in the generated taxonomy.

2. Objective-function based explanations: Duong et al. (2019) propose a loss function that minimizes the number of connections between communities, and in doing so, the nodes in a community are embedded closer to each other. Explainability comes in the form of each dimension defining the degree of membership to a community.

3. Explanations per embedding dimension:

   - Gogoglou et al. (2019) define an interpretability score for each dimension in a node embedding. It measures how well a dimension is associated with a subgroup of nodes.
   - Khoshraftar et al. (2021) define the interpretability of an embedding algorithm as the extent to which its embedding dimensions can represent "important" nodes in a category with extreme values, where "importance" is based on centrality measures.

4. Node-property explanations: Dalmia et al. (2018) and Bonner et al. (2019) study how good an embedding algorithm is at explaining certain node properties such as node degree or node centrality. This is akin to what we have referred to as "sense features".

5. Embedding-based explanations: Rossi et al. (2018) use features akin to what we refer to as "sense features" as base features, to which relational feature operators are applied to generate embeddings. Explainability comes in the form of examining which relational operators were used to generate the embeddings.

Compared with existing literature, our proposed method XM is independent of any particular downstream task, can be used to extend any existing embedding algorithm, and works with any set of sense features. It also provides explanations on the node- and dimension-level.

## 3   Explaining embeddings

### 3.1   Sense-making

In order to explain each embedding dimension, we define a set of human-understandable node features, called "sense" features. We use global and local properties of the network as our "sense" features. Following Ghasemian et al. (2020), we look at the following 15 features: degree, weighted degree (if the edges are weighted), clustering coefficient (Barabási, 2016), average of the personalized PageRank vector, standard deviation of the personalized PageRank vector, average degree of the neighboring nodes, average clustering coefficient of neighboring nodes, number of edges in the ego net, structural hole constraint (Burt, 2004), betweenness centrality (Freeman, 1977), eccentricity, PageRank, degree centrality, Katz centrality (Katz, 1953) and eigenvector centrality (Newman, 2010). These features are well-established, easily computable, and provide a comprehensive glossary for graph analysis providing insights into the overall structure of the graph (e.g., average degree, clustering coefficient, number of triangles) as well as detailed information at the individual node level (e.g., degree, betweenness centrality). Since some of these features are highly correlated, we choose to focus on the following subset of them throughout this work: degree, clustering coefficient, standard deviation of the personalized PageRank, average degree of the neighboring nodes, average clustering coefficient of neighboring nodes, eccentricity and Katz centrality.

Given an embedding vector for a node $k$, $\vec{y}_k \in \mathbb{R}^{d \times 1}$ and its corresponding sense feature vector $\vec{f}_k \in \mathbb{R}^{f \times 1}$ (where $d$ is the number of dimensions of the embedding space and $f$ is the number of sense features used, each being normalized between 0 and 1), we compute the element-wise similarity between the embedding

vector for a node and its corresponding sense feature vector as follows:

$$E_k = \frac{\vec{y}_k \otimes \vec{f}_k^T}{\|\vec{y}_k\|\|\vec{f}_k\|} \tag{1}$$

We call $E \in \mathbb{R}^{d \times f}$ the *Explain* matrix (we drop the subscript $k$ corresponding to node $k$, for brevity). Normalized to a range of 0 and 1, each row $i$ of the Explain matrix corresponds to a dimension in the embedding space and each column $j$ corresponds to a sense feature. The value $E_{ij}$ corresponds to how much the dimension $i$ is defined by the sense feature $j$. This matrix $E$, helps us to investigate the linear relationships between the sense feature vectors and the embedding vector.

### 3.1.1 Real world example

We look at the Karate Club Network for a concrete real-world example. We chose this network for brevity and pedagogy and discuss larger networks in the Results section. Figure 2(A) shows the original Karate Club Network (Zachary, 1977). We embed the Karate Club graph into 16 dimensions using DGI Velickovic et al. (2019). We then visualize these embeddings by projecting them into 2 dimensions using UMAP (McInnes et al., 2018) (shown in Figure 2(B)). For a detailed node-wise view, we look at the president of the club (node 34), the instructor (node 1), and a random student (node 12). We use Equation (1) to compute the Explain matrix for each node. Figure 2(C) corresponds to the Explain matrix for the instructor (node 1), Figure 2(D) corresponds to the random student (node 12), and Figure 2(E) to the president (node 34). Examining the Explain matrices for nodes 1 and 34 (Figures 2(C) and (D)), we see similar sense features (like degree, the standard deviation of the personalized PageRank, Katz centrality and average neighbor clustering) stand out and are explained most by dimensions 14 and 15 (note that the specific dimensions are not constant). Also, observe that these two nodes are placed close together in the embedding space (low-dimensional visualization in Figure 2(B)). We also see that the Explain matrix for node 12 (Figure 2(D)) is different, and has average neighbor degree and eccentricity as its distinguishing features.

### 3.1.2 Choice of Sense Features

The set of features defined earlier is not exhaustive but was chosen to ensure consistency across different datasets and are well-established in domains that represent data as graphs (Albert & Barabási, 2002; Newman, 2003). For datasets that already have inherent node features, we deliberately exclude them and instead use sense features to enable fair comparisons and analysis across various datasets.

Here, we explore the impact of using different sense features. Observe how the sense features defined above are primarily structural, meaning they capture the neighborhood structure of nodes without considering their specific positions within the network. Consequently, nodes with similar neighborhood structures in the same network will have similar sense feature vectors. The explanations produced by these sense features would therefore be more role-based, i.e. would emphasize the role of a node in the network (e.g. gatekeeper node or hubs).

However, if the explanations one is looking for are more positional in nature,[2] i.e., explanations that focus on a node's location relative to others in the network, structural sense features might not be as effective. To demonstrate the differences in sense features, as well as what the Explain matrix contains, we define a set of "positional" sense features. We follow You et al. (2019) in defining the concept of anchor nodes and use simple features such as the number of hops to the anchor node as our positional sense features. As before, we embed the Karate club network using DGI into 16 dimensions and generate the Explain matrices for each node in the graph following Equation (1).

Figure 3 visualizes the Explain matrices for nodes 1 (instructor), 12 (random node), and 34 (president). We select four nodes as anchor nodes. Nodes 3 and 9 behave as bridge nodes between the communities of the instructor and the president. Nodes 5 and 33 only have connections to the instructor and president communities, respectively. We compute the features for each node $i$ in the graph, as the number of hops from

---

[2]Positional explanations look at the role of a node instead of its community.

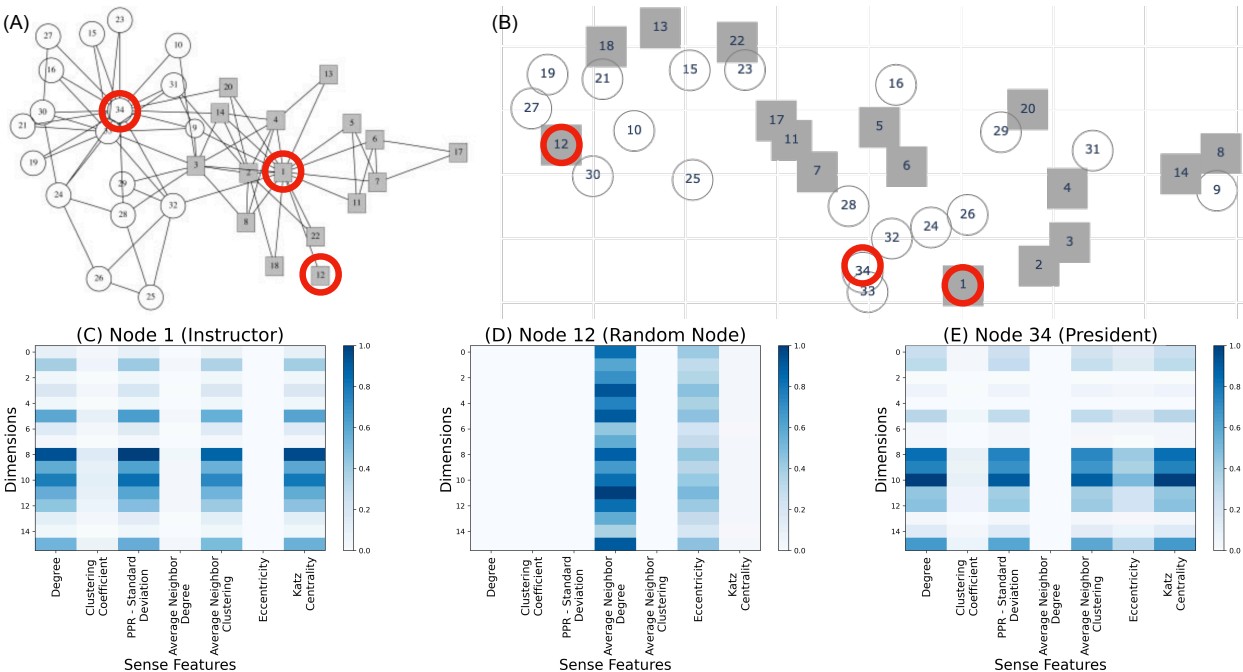

Figure 2: **(A)** Karate Club Network (Zachary, 1977). **(B)** We embed the Karate Club network into 16 dimensions using DGI and visualize it in 2 dimensions using UMAP (McInnes et al., 2018). **(C)** Explain matrix for the instructor (node 1). **(D)** Explain matrix for a random student node (node 12). **(E)** Explain matrix for the president (node 34). Examining the Explain matrices for each of these nodes gives us an understanding of their placement in the embeddings space. Observe how features such as degree, personalized page rank etc., stand out for nodes 1 and 34 (subplots **(C)** and **(E)**) and features like average neighbor degree stands out for the random student node (subplot **(D)**).

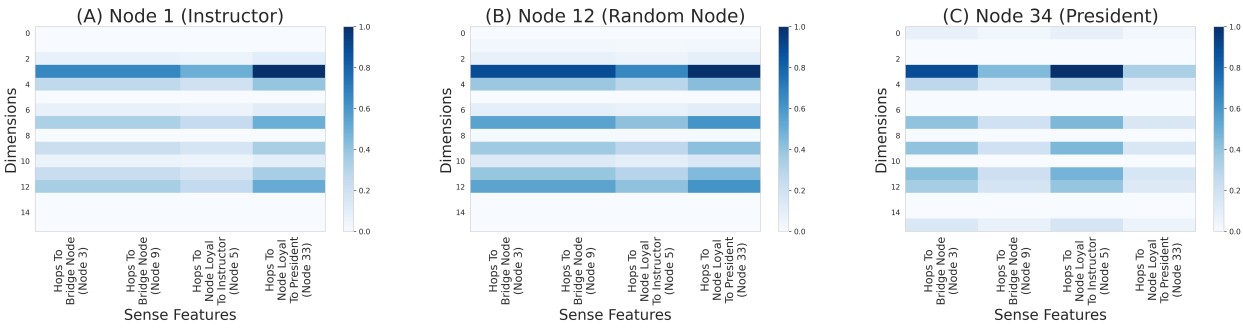

Figure 3: Visualization of Explain matrices for three nodes in the Karate Club network - node 1 (instructor, A), node 12 (random node, B), and node 34 (president, C) - based on positional sense features defined as number of hops to 4 anchor nodes (nodes 3, 5, 9 and 33). The matrices highlight the similarity between nodes 1 and 12 due to their relative positions in the network and distances from the four anchor nodes, in contrast to the structural feature-based similarities observed in Figure 2

the given node $i$ to these four anchor nodes. While in Figure 2, nodes 1 and 34 displayed similar Explain matrices due to their comparable structural features, Figure 3 reveals greater similarity between nodes 1 and 12. This similarity is due to their relative positions in the graph and their respective distances from each of the anchor nodes. This contrast illustrates how positional sense features can yield different insights compared to structural features.

### 3.2 Our Proposed Framework: XM (eXplain eMbedding)

#### 3.2.1 Quantifying the Quality of Explanations

Given a method for generating explanations, we quantify the quality of these explanations. The (potentially) large number of dimensions and sense features drives us to use heatmaps to visualize their relationships. Figure 2 shows the heatmap visualization of the Explain matrix. Notice that a noisy Explain matrix is hard to interpret, but sparser Explain matrices with each dimension being defined by unique sets of sense features are easier to interpret.

Since this translates to a denoised Explain matrix, we draw inspiration from the work on image denoising using nuclear norm minimization [Gu et al. (2014), Xu et al. (2017)]. In this context, the nuclear norm serves as a quantitative metric, with lower nuclear norms indicating coherent and interpretable explanations.

#### 3.2.2 The Importance of Rank: Interpreting Rank Reduction through Nuclear Norms

Given a matrix of rank $r$, one can reconstruct it using $r$ rank-1 matrices. The nuclear norm of a matrix is the tightest convex bound of the rank function over the unit ball of matrices with bounded spectral norm.[3] Thus, the difference in nuclear norms of two matrices gives us a lower bound on how many more rank-1 matrices are needed to reconstruct the matrices.

For example, SDNE on the EU Email dataset (with $d = 128$) produces an Explain matrix whose nuclear norm is $\simeq 8 \pm 0.97$ (see Figure 12). However, its Explain matrix has a nuclear norm of $\simeq 1.8 \pm 0.26$ when the sparsity constraint is used, $\simeq 2.1 \pm 0.29$ when the orthogonality constraint is used, and $\simeq 1.6 \pm 0.39$ when both constraints are used. This means that the Explain matrix for the EU Email dataset can be approximated with two rank-1 matrices when the sparsity and orthogonality constraints are used, instead of eight rank-1 matrices with no constraints used. (The error bars in these measurements represent the standard error, underscoring the statistical significance of the improvements observed.)

#### 3.2.3 Augmenting Embeddings

It is important to avoid a perfect one-to-one mapping between the embedding dimensions and the sense features. If a one-to-one correspondence is imposed, then the network information beyond that contained in the simple node sense features is lost while creating the embeddings. Such embeddings would not capture higher-order network properties that contribute to downstream tasks (Ghasemian et al., 2020), resulting in poor performance. Using sense features themselves as embedding vectors or performing dimensionality reduction on a large set of sense features suffers from the same issue of not capturing higher-order network properties. To that end, we propose XM, a framework for augmenting existing embedding algorithms that includes sense feature information, as well as information that the underlying algorithm captures.

XM incorporates sense features and achieves a less noisy Explain matrix by imposing a sparsity constraint along the columns of the Explain matrix. This aims to zero-out features with small contributions to the definition of a dimension. Following recent advances in dimensional contrastive learning by Nguyen et al. (2023), XM also imposes an orthogonality constraint along the rows of the Explain matrix which aims to have embedding dimensions that are defined by different sets of sense features. XM uses the two constraints, sparsity and orthogonality, instead of directly minimizing the nuclear norm for added control over the objective function (see section 4.5). The loss terms for the two constraints can be written as follows.

$$L_{sparse} = \|E_k[:,j]\|_1 \qquad \forall j \in columns \qquad (2)$$

$$L_{ortho} = \|E_k[i,:]E_k^T[j,:]\|_2 \qquad \forall i \in rows\ \forall j \in columns \qquad (3)$$

These two additional loss terms are then added to the objective functions of existing embedding algorithms. As an example, the objective for SDNE consists of 3 loss terms - a first order loss term, a reconstruction

---

[3]Note that the rank of a matrix is a non-convex function, which makes optimization problems involving the rank hard to solve. The nuclear norm serves as a convex proxy for the rank.

error term and a regularization term:

$$L_{SDNE} = \alpha L_{first} + \beta L_{recon} + \nu L_{reg} \tag{4}$$

where $\alpha$, $\beta$, and $\nu$ are tunable hyperparameters.

We augment this objective as follows :

$$L_{SDNE} = \alpha L_1 + \beta L_2 + \nu L_{reg} + \gamma L_{Sparse} + \delta L_{Ortho} \tag{5}$$

which we refer to as SDNE+XM. Here, $\gamma$ and $\delta$ are hyperparameters for the sparsity and orthogonality constraints, respectively.

### 3.2.4 Nuclear Norm and Matrix Entropy

For a given node, assume there exists an embedding $y^*$ (we drop the vector notation $\vec{y}$ for brevity), which is the optimal embedding. Let $y^{xm}$ be the embedding for the same node found by XM. Recall that we define the Explain matrix $E$ as:

$$E^* = \frac{y^* \otimes f^T}{\|y^*\|\|f\|} \tag{6}$$

$$E^{xm} = \frac{y^{xm} \otimes f^T}{\|y^{xm}\|\|f\|} \tag{7}$$

Let

$$A = E^* E^{*T} \tag{8}$$

$$B = E^{xm} E^{xm^T} \tag{9}$$

Observe that $A$ and $B$ are now symmetric matrices and therefore have an orthonormal basis of eigenvectors with real eigenvalues.

We then have

$$A = U\Sigma U^T \qquad \text{where } U \text{is orthonormal and } \Sigma \text{ is diagonal} \tag{10}$$

$$z^T A z = z^T \ U\Sigma U^T \ z \qquad \forall z \geq 0 \in \mathbb{R}^n \tag{11}$$

$$z^T A z = \sum_{i=1}^{n} \Sigma_{ii}(U^T z)_i^2 \tag{12}$$

$$z^T A z \geq 0 \tag{13}$$

This shows that $A$ is a positive semi-definite (PSD) matrix. The same proof applies for the matrix $B$. Given that $A$ and $B$ are PSD, Yu (2013) and Bach (2022) show that when the Von Neumann entropy of a PSD matrix is defined as

$$H(A) = -tr[A \ log \ A] \tag{14}$$

and its associated Bregman Divergence is defined as

$$D(A\|B) = tr[A(log \ A - log \ B)] - tr(A) + tr(B) \tag{15}$$

the following extension of Pinsker's inequality holds true:

$$D(A\|B) \geq \frac{1}{2}\|A - B\|_*^2 \tag{16}$$

Table 2: Embedding algorithms used. Classification of these algorithms was adopted from Chami et al. Chami et al. (2022), where we pick two algorithms that allow for a node feature matrix, and two algorithms that do not. Providing sense features as node attributes is a simple method of incorporating the sense features into the embeddings. XM variants outperform the original versions in terms of nuclear norms.

| Algorithm | Node Features | Sub Type |
|---|---|---|
| SDNE Wang et al. (2016) | No ($X = I$) | Autoencoder |
| LINE Tang et al. (2015) | No ($X = I$) | Skip Gram |
| DGI Velickovic et al. (2019) | Yes ($X \neq I$) | Message Passing |
| GMI Peng et al. (2020) | Yes ($X \neq I$) | Message Passing |

We also know that the nuclear norm is an upper bound on the $L_2$ norm, i.e.

$$\|A\|_* \geq \|A\|_2 \tag{17}$$

This gives us

$$D(A\|B) \geq \frac{1}{2}\|A - B\|_*^2 \geq \frac{1}{2}\|A - B\|_2^2 \tag{18}$$

$$D(A\|B) \geq \frac{1}{2}[\|A\|_2^2 - 2\|A\|_2\|B\|_2 + \|B\|_2^2] \tag{19}$$

Recall that

$$B = E^{xm}E^{xm^T} \tag{20}$$

which is explicitly minimized as the orthogonality constraint in the XM framework. Notice that the $\|A\|_2$ term is constant since it is the optimal solution. Thus, by minimizing the $L_2$ norm of $B$, we reduce the lower bound in the entropy between an optimal explanation $E^*$ and the explanation $E^{xm}$ found by XM. The relationship between entropy and prediction has been well known for a long time (Shannon (1951)). More recently, the impact of entropy reduction on machine learning explanations has gained importance. Amanova et al. (2024) present a method for image explainability that identifies input features affecting the model's output entropy. Reducing prediction entropy can improve explanations by making the model's decisions more confident. Press et al. (2024) and Wang et al. (2020) demonstrate that entropy minimization during test time optimizes model confidence and accuracy, revealing a strong correlation between lower output entropy and higher accuracy. As entropy decreases, model predictions become more decisive, leading to clearer and more accurate explanations of the model's behavior.

## 4 Experiments

In order to demonstrate our method, we apply our augmentation to 4 unsupervised embedding algorithms across 6 different networks. We look at the taxonomy of machine learning on graphs by Chami et al. (2022) and pick two algorithms from the branch where $X = I$, (i.e. these algorithms do not incorporate node attributes) and two algorithms from the branch where $X \neq I$ (i.e. these algorithms do allow node attributes). We choose these algorithms because providing sense features as node attributes is a simple and straightforward method to incorporate the sense features into the embeddings. However, as we shown below, we achieve lower nuclear norms by augmenting the objective function.

Table 2 shows the algorithms selected and Table 3 summarizes the datasets used. These have been chosen to span a range of average degrees and clustering coefficients.

We start by visualizing the Explain matrices for the Karate Club network embedded using DGI. We then discuss quantitative metrics for the networks listed in Table 3, embedded using each of the four embedding algorithms from Table 2 and their XM variants. Following this, we dive into a case study on the largest

Table 3: Real-world networks used in our experiments. $\langle k \rangle$ refers to the average degree, $\sigma_k$ refers to the standard deviation of the degree, $r$ refers to the degree assortativity and $c$ refers to the average clustering coefficient. These networks were picked to span a range of average degrees and clustering coefficients.

| Network | Nodes | Edges | $\langle k \rangle$ | $\sigma_k$ | $r$ | $c$ | Transitivity |
|---|---|---|---|---|---|---|---|
| EU Email Leskovec et al. (2007) | 986 | 16687 | 33.84 | 37.81 | -0.01 | 0.40 | 0.26 |
| US Airport Zhu et al. (2021) | 1186 | 13597 | 22.92 | 40.49 | 0.03 | 0.50 | 0.42 |
| Squirrel Rozemberczki et al. (2021) | 5201 | 198493 | 76.32 | 161.45 | -0.23 | 0.42 | 0.34 |
| Citeseer Bollacker et al. (1998) | 3327 | 4676 | 2.81 | 3.38 | 0.05 | 0.14 | 0.13 |
| FB15K-237 Toutanova & Chen (2015) | 14951 | 261581 | 34.99 | 111.43 | -0.10 | 0.213 | 0.02 |
| PubMed Roberts (2001) | 19717 | 44327 | 4.49 | 7.43 | -0.04 | 0.06 | 0.05 |

network in our datasets, PubMed, to demonstrate a practical application of our approach and compare our results against other explanation methods. Finally, we present an ablation study that explores the impact of different constraints of XM.

## 4.1 Explanations - Visual Evaluation

Figure 4, shows the Explain matrix for the Karate Club network. As in Figure 2 (C)(D) and (E), the three subplots of Figure 4 show the Explain matrices for the instructor node (node 1, Fig 4(A)), a random student node (node 12, Fig 4(B)) and the president node (node 34, Fig 4(C)). To stay consistent across experiments, these plots were generated using 16-dimensional embeddings that were generated from DGI+XM. Observe that the Explain matrices shown in Figure 4(A)(B) and (C), while highlighting similar sense features, are sparser than the respective plots in Figure 2(C)(D) and (E) and each sense feature is defined by fewer dimensions, i.e. the dimensions are orthogonal in their explanations when compared to the standard DGI.

For each of our six real world datasets, we also compute an Explain matrix for the entire graph by aggregating the node embeddings into a graph-level embedding, using a mean readout. Similarly, we compute the mean of the sense feature matrix. Using Equation 1, we derive a graph-level Explain matrix, as shown in Figure 5. For brevity, we only show results computed using DGI+XM and 32 dimensional embeddings. In particular, Katz centrality emerges as the most prominent feature for networks such as Citeseer and Pubmed, which is intuitively consistent with the nature of citation networks, as Katz centrality measures the degree of influence of nodes within a network Katz (1953). In contrast, the salient features for the EU Email data are clustering and average neighbor clustering, which is consistent with social networks.

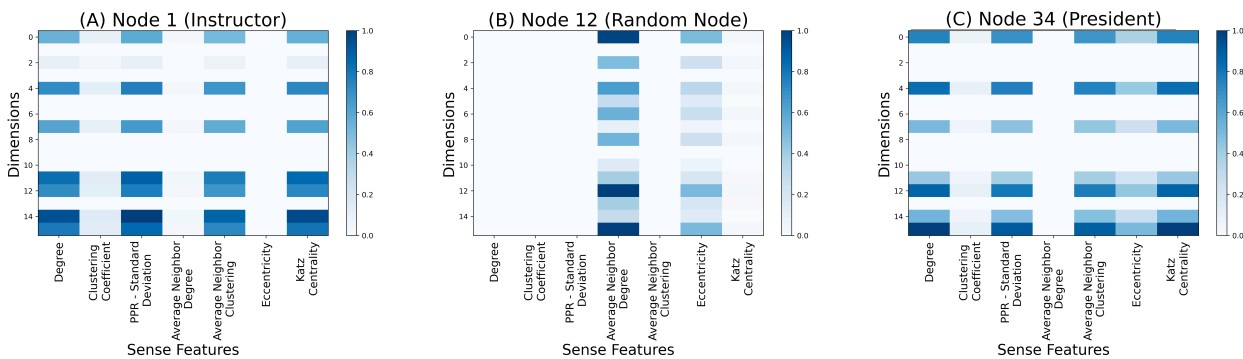

Figure 4: Explain matrices for the Karate Club network (Zachary, 1977) generated using DGI+XM for (A) the instructor (node 1), (B) a random student (node 12), and (C) the president (node 34). Compare each of (A), (B), and (C) to Figure 2(C),(D), and (E), respectively. Observe that the Explain matrices in this figure (generated by DGI+XM) are sparser and each sense feature is explained by fewer dimensions, when compared to the Explain matrices from Figure 2 (generated by the standard DGI).

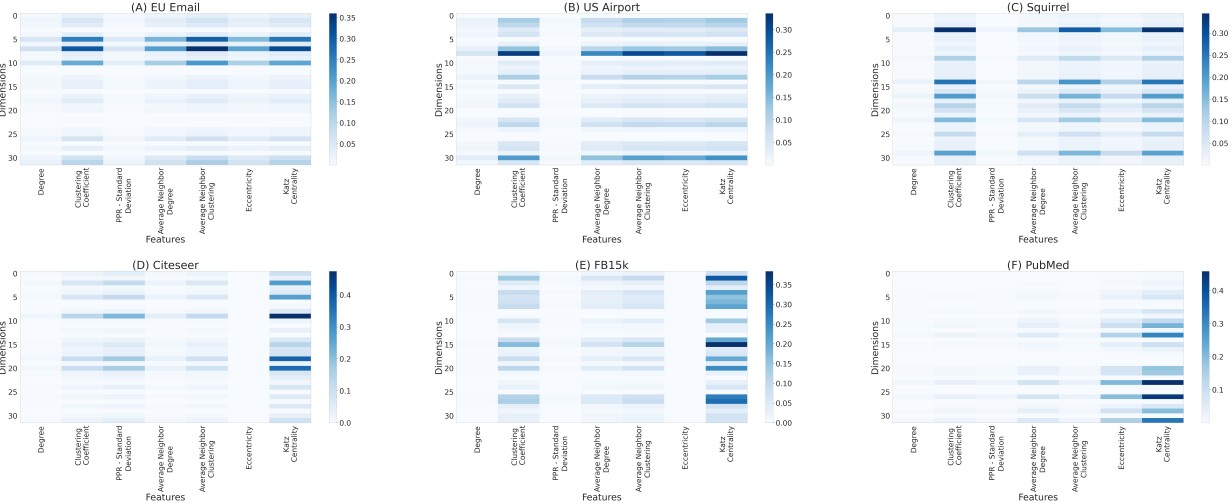

Figure 5: Graph-level Explain matrices for six real-world datasets, based on a mean readout of all node embeddings computed using DGI+XM and mean sense features. The prominent features across different networks are highlighted, with Katz centrality emerging as the dominant feature in citation networks like Citeseer and Pubmed, while clustering and average neighbor clustering stand out in social networks like the EU Email network.

## 4.2 Explanations - Quantitative Evaluation

We examine the nuclear norms of the Explain matrices as a quantitative metric to evaluate and compare the standard and XM variants of the algorithms. For each dataset, we use each of the four algorithms shown in Table 2 and compute the mean of the nuclear norm of the Explain matrices across all nodes in the dataset. We embed every dataset into 128 dimensions for consistency across datasets and algorithms and pass in sense features as node attributes when running DGI and GMI.

As discussed in Section 3.2.2, lower nuclear norms result in sparser Explain matrices, meaning fewer rank-1 matrices are required to reconstruct the given matrix. This sparsity is indicative of interpretable explanations. Figure 6 demonstrates that for each node of a graph, the nuclear norm of its Explain matrix is reduced when using SDNE+XM versus SDNE. The distribution of nuclear norm values shifts leftward (i.e. lower), indicating that the XM framework successfully reduces nuclear norms (thereby, ranks) across all datasets. Similar results for other algorithms can be found in the appendix in Figures 13, 14, and 15

## 4.3 Link Prediction Performance

By adding additional loss terms to the objective function, we modify the embeddings themselves to generate a less noisy Explain matrix. We use link prediction as our downstream task and sample an equal number of positive and negative edges from the graph. We use 60% of the edges as training data and the remaining as test data. The embeddings are passed through a simple 2 layer fully connected neural network. We repeat the entire process 3 times to create a three-fold cross-validation setup. We use 128-dimensional embeddings across all algorithms and networks for consistency.

Figure 7 depicts the results with nuclear norm (averaged across all nodes in the graph) shown on the $y-$axis and the AUC shown on the $x-$axis with error bars denoting the standard error. We observe that across all algorithms, AUC scores are comparable to the original version, with the corresponding nuclear norms being lower leading to denoised Explain matrices.

We also examine differences in runtime due to the additional constraints in Figure 8. More specifically, we examine the time (in seconds) per epoch for each of the 4 algorithms we test for the EU Email network. We observe that the XM variants are comparable in runtime to their original versions. Results shown are

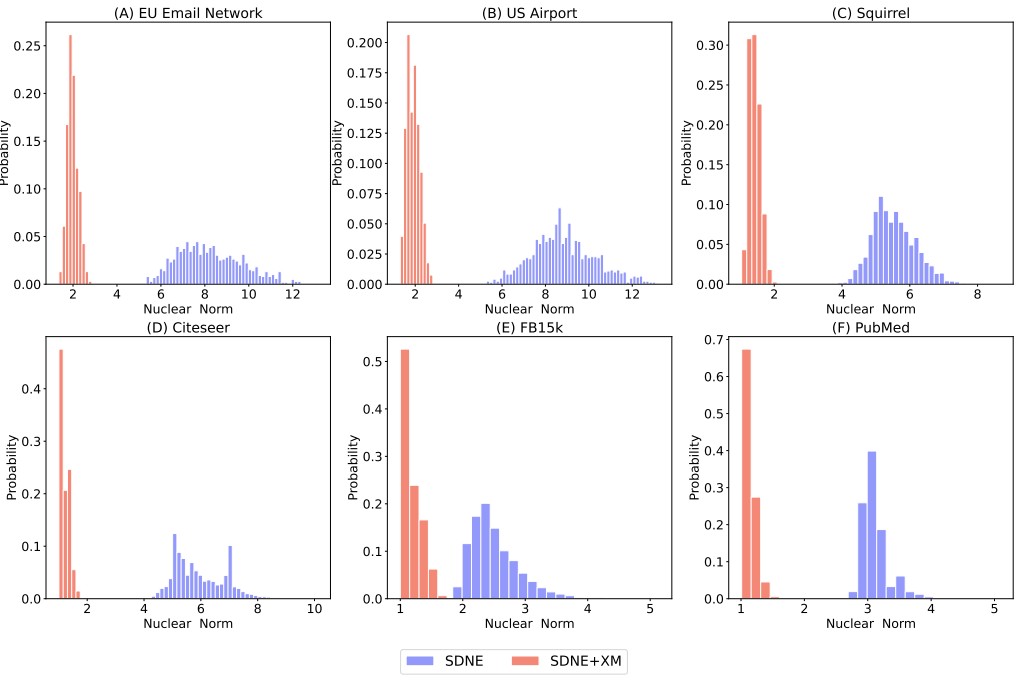

Figure 6: Distribution of nuclear norms of the Explain matrices for each node for SDNE and SDNE+XM. Observe how the distribution of SDNE+XM is shifted to the left with a lower mean across each of the 6 datasets. See Section A.2 in the appendix for results on the other methods.

averaged across five runs with error bars showing standard error. Similar runtime comparisons for other datasets are shown in the appendix (Figures 16 through 20).

## 4.4   Case Study - PubMed Network

To illustrate a concrete use case, we focus on the PubMed network and use DGI as the embedding algorithm. The nodes in PubMed are classified into three categories - Diabetes Mellitus, Experimental (Class 0), Diabetes Mellitus Type 1 (Class 1) and Diabetes Mellitus Type 2 (Class 2). It is important to note that DGI does not take these class labels into account while training. For this study, we embed the network into 32 dimensions using DGI+XM for brevity. We then randomly select eight nodes, ensuring a variety of connections and degrees, including nodes that are both connected and disconnected, with both matching and differing class labels. Details of these nodes are presented in Table 4. The resulting Explain matrices for these eight nodes are presented in Figure 9. Each matrix illustrates the key sense features responsible for positioning these nodes within the embedding space, allowing for a clear interpretation of the role each node and feature plays.

We next compare the explanations generated by DGI and its augmented variant, DGI+XM, focusing on the importance of each dimension across nodes. Unlike prior experiments that utilized structural sense features, we employ PubMed's 500 inherent node features to showcase an application with non-structural features. Specifically, we compute the average of each row in the Explain matrix, where this average reflects the relative significance of each dimension in explaining the inherent features of the nodes in PubMed.

In Figure 10, the y-axis represents the mean dimensional importance for each of the 32 dimensions shown on the x-axis. The top row of the figure illustrates the results from DGI+XM, while the bottom row shows results from the standard DGI variant. Notably, DGI+XM provides clearer and interpretable results. For example, for connected nodes within the same class (first column), the important dimensions stand out more distinctly in DGI+XM compared to the noisier and less discernible results from DGI. These differences

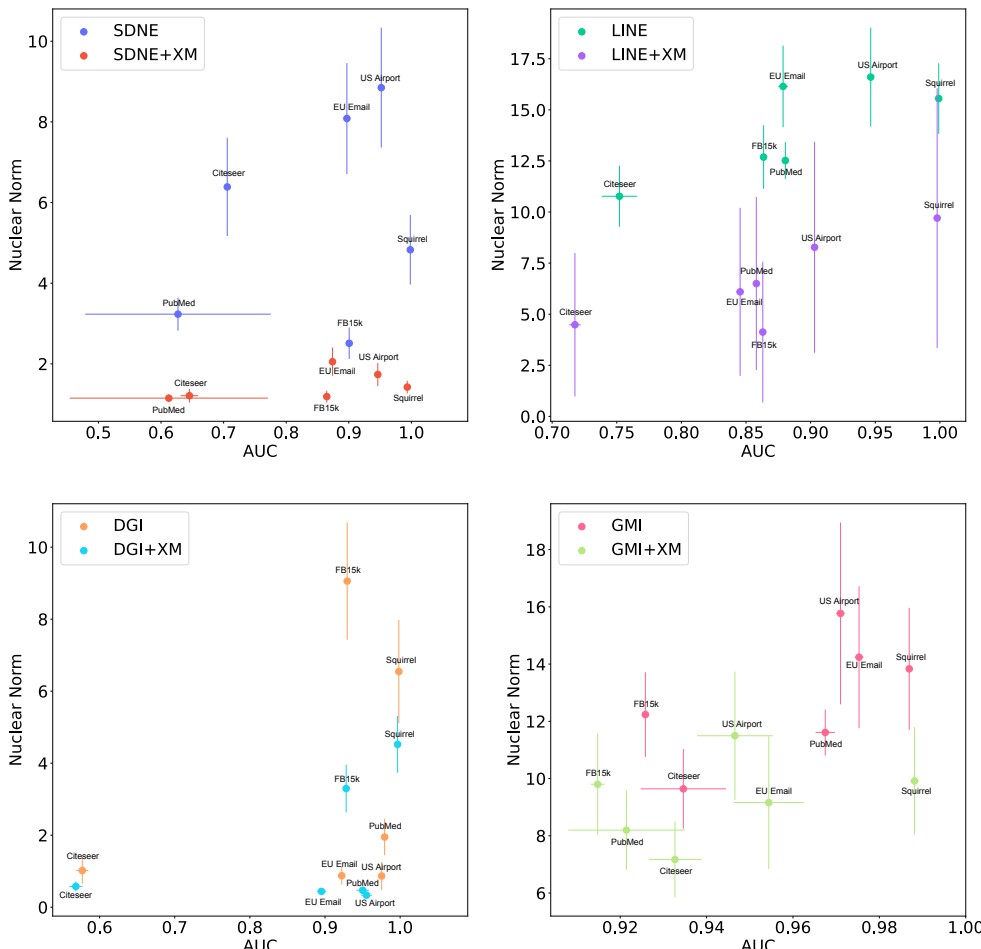

Figure 7: AUC scores for link prediction across 4 embedding algorithms - SDNE, LINE, DGI, GMI - shown in each subplot, on the 6 networks defined in Table 3. Error bars show standard error. AUC scores for the XM variants are comparable to the original algorithms with nuclear norms of the XM variants being lower. The XM variants also follow the original algorithms in terms of standard error, for example, SDNE has large variance in AUC for the PubMed dataset, which can also be seen for SDNE+XM. Results are from a three-fold cross validation experiment with 128 dimensional embeddings.

become even more pronounced in other scenarios, such as when nodes belong to different classes or are disconnected. As another example, with DGI+XM, dimensions 14, 27 and 28 are highlighted as particularly important for nodes in Class 2 across nodes, while similar insights are difficult to obtain from the standard DGI approach due to its noisier nature, even considering non-identifiability issues.

We now compare our approach with two widely-used graph explanation methods: GNNExplainer (Ying et al., 2019) and GraphMask Explainer (Schlichtkrull et al., 2021). Both of these methods focus on explaining the decisions of a trained model, particularly in node classification tasks. In contrast, XM is not designed to explain specific classification outcomes or generate subgraphs for explanations. For the sake of comparison however, we analyze the Explain matrix of a node by averaging across its columns to identify the most important features.

For GNNExplainer and GraphMask Explainer, we train a 2-layer GCN with 32 hidden units each for node classification and apply the explainers post-hoc to generate node feature importance scores. It is important to note that GNNExplainer and Graph Mask Explainer specifically target explaining node classification in

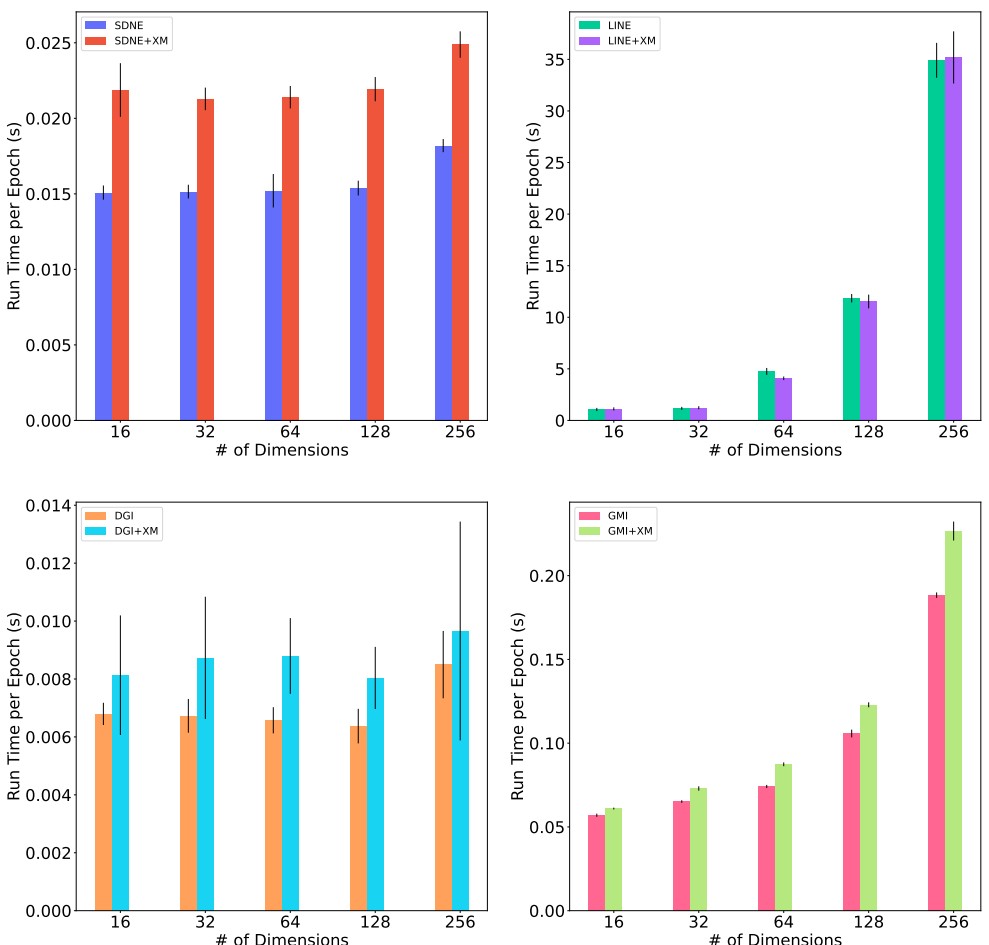

Figure 8: EU email network: runtimes per epoch (in seconds) for SDNE, LINE, DGI, GMI, and their corresponding XM variants. Results are averaged across 5 runs. Observe that the runtimes per epoch are comparable to the original versions. Error bars show standard error. See Section A.3 in the Appendix for the resutls on other networks.

| Node | Degree | Clustering Coefficient | # Triangles | Class | Paper |
|------|--------|------------------------|-------------|-------|-------|
| A | 1 | 0.000 | 0 | 2 | Malhotra et al. (1992) |
| B | 19 | 0.046 | 8 | 2 | Petrovsky et al. (2002) |
| C | 171 | 0.006 | 96 | 1 | Group et al. (1998) |
| D | 33 | 0.075 | 40 | 0 | Obrosova et al. (2005) |
| E | 10 | 0.044 | 2 | 2 | Støy et al. (2008) |
| F | 5 | 0.100 | 1 | 2 | Cline et al. (1994) |
| G | 1 | 0.000 | 0 | 0 | Sima & Sugimoto (1999) |
| H | 1 | 0.000 | 0 | 2 | Badenhoop & Boehm (2004) |

Table 4: Description of randomly selected nodes for the PubMed case study. Here class 0 corresponds to papers regarding Experimental treatments for Diabetes Mellitus, class 1 about Diabetes Mellitus Type 1 and classs 2 about Diabetes Mellitus Type 2

the context of a trained model, whereas XM (specifically DGI+XM as used before) remains task-agnostic and focuses on explaining embedding dimensions.

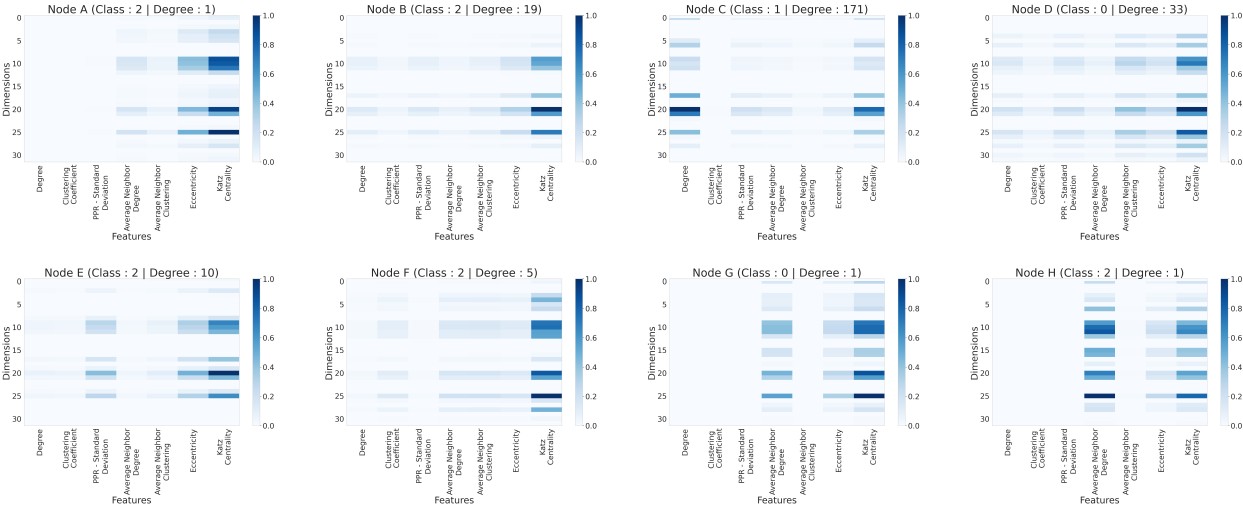

Figure 9: This figure illustrates the Explain matrices for eight randomly selected nodes from the PubMed network, highlighting the sense features responsible for their positioning within the embedding space. Each row corresponds to a dimension of the embedding, while each column represents a human-understandable feature.

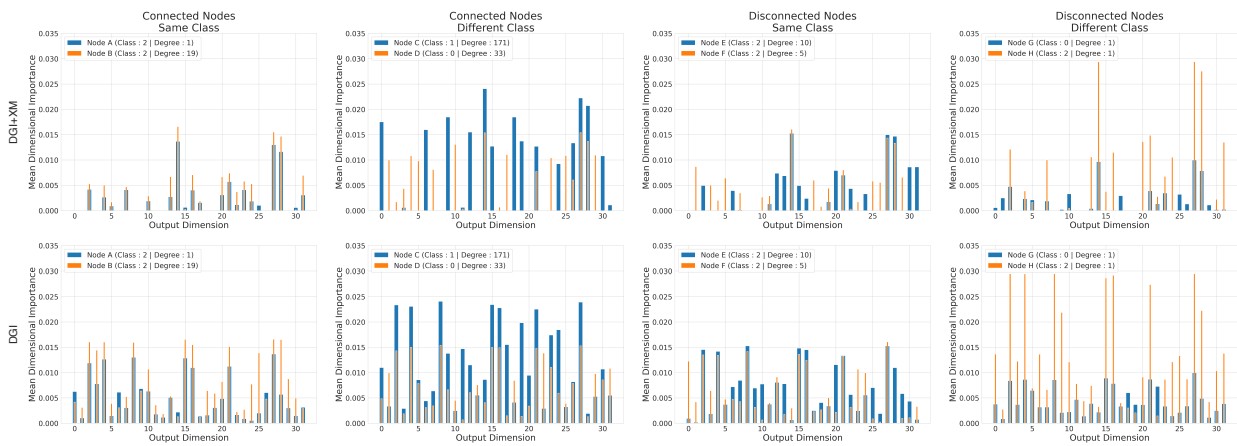

Figure 10: Comparison of dimensional importance, computed as the row wise mean of the Explain matrices, for randomly selected nodes in the PubMed network using DGI+XM (top row) and DGI (bottom row). Observe the clarity of the explanations provided by DGI+XM, particularly in identifying important dimensions for nodes with the same or different class labels and varying connection statuses as opposed to noisier explanations provided by the standard DGI variant.

For the 8 nodes previously discussed, we identify the top 25 features that each explainer model highlights as significant. To quantify the similarity between these feature sets, we compute the Jaccard similarity across the top 25 features identified by each method, as illustrated in Figure 11. Interestingly, despite XM not being tailored for feature importance, it still identifies many of the same features highlighted by GNNExplainer and Graph Mask Explainer in their top 25. We also observe a higher degree of alignment between XM and GNNExplainer, potentially due to GNNExplainer's explicit focus on maximizing mutual information between feature subsets and the original feature set.

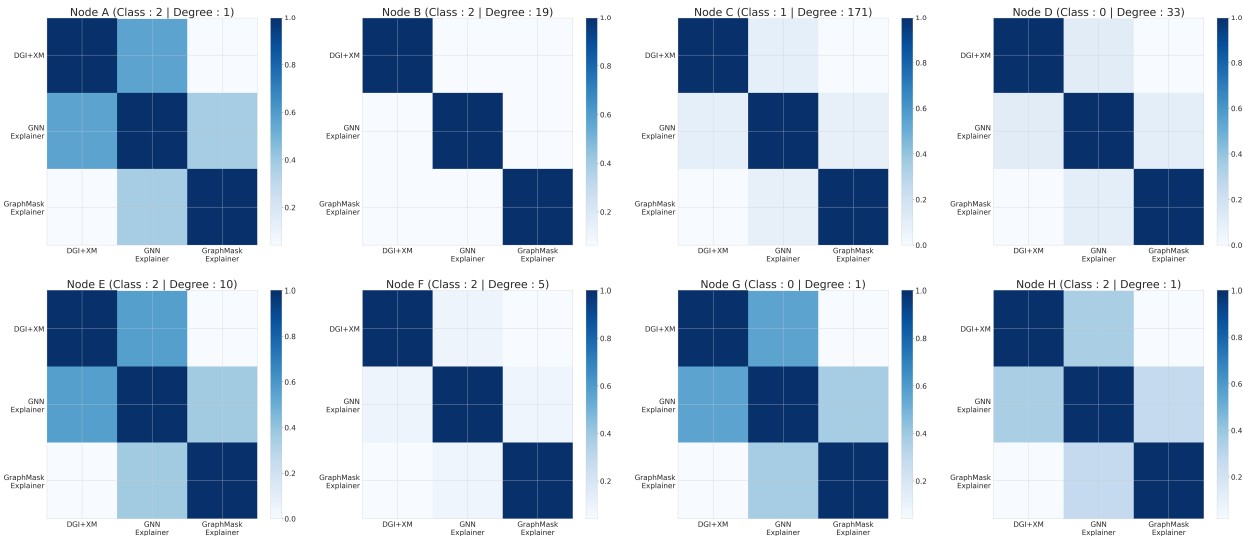

Figure 11: Comparison of the top 25 features identified as important for the task of node classification by XM, GNNExplainer, and GraphMask Explainer across 8 randomly selected nodes, representing different classes and degrees. To quantify the similarity between these feature sets, we compute the Jaccard similarity between the list of features highlighted by each method. Despite XM not being specifically designed for explaining node classification outcomes, it still highlights many of the same features identified by GNNExplainer and GraphMask Explainer, with higher alignment with GNNExplainer than GraphMask Explainer.

## 4.5 Ablation Study

We add two constraints, orthogonality and sparsity, while computing the augmented node embeddings. We study how each contributes to the reduction of nuclear norms. Figure 12 shows nuclear norms for each network embedded in 128 dimensions using each of the four embeddings algorithms, averaged across five runs. We see that the orthogonality constraint alone reduces the nuclear norm more than the sparsity constraint alone. We see that across all algorithms, using both constraints together achieves the lowest nuclear norm.

## 5 Discussion

The XM framework generates explanations at the granularity of each dimension for individual nodes. While this level of detail is valuable for examining specific nodes, deriving a comprehensive understanding of a large network requires further analysis of multiple Explain matrices.

As is common in works on explainability, the use of the XM framework leads to a marginal reduction in performance scores for downstream tasks. However, note that the reduction in performance can be controlled through the choice of hyperparameters. For this study, we chose to prioritize the quantitative value of the nuclear norm to obtain the largest possible reduction. In actual use, the practitioner can balance the amount of denoising of the Explain matrices to the desired level of performance in downstream tasks by setting the hyperparameters accordingly. (Our choices for the hyperparameters are available in our online code repository at https://github.com/zohairshafi/ExplainingNodeEmbeddings)

The effectiveness of XM is also influenced by the choice of sense features. As discussed in Section 3.1.2, different sense features can yield different results. Domain knowledge plays a role in selecting the appropriate features, depending on the specific goals of the explanation task. We chose to use structural sense features throughout this work because they provide a reasonable and human-interpretable basis for understanding graphs across various domains, such as sociology and biology, using features like node degree and clustering coefficient (Albert & Barabási, 2002; Newman, 2003).

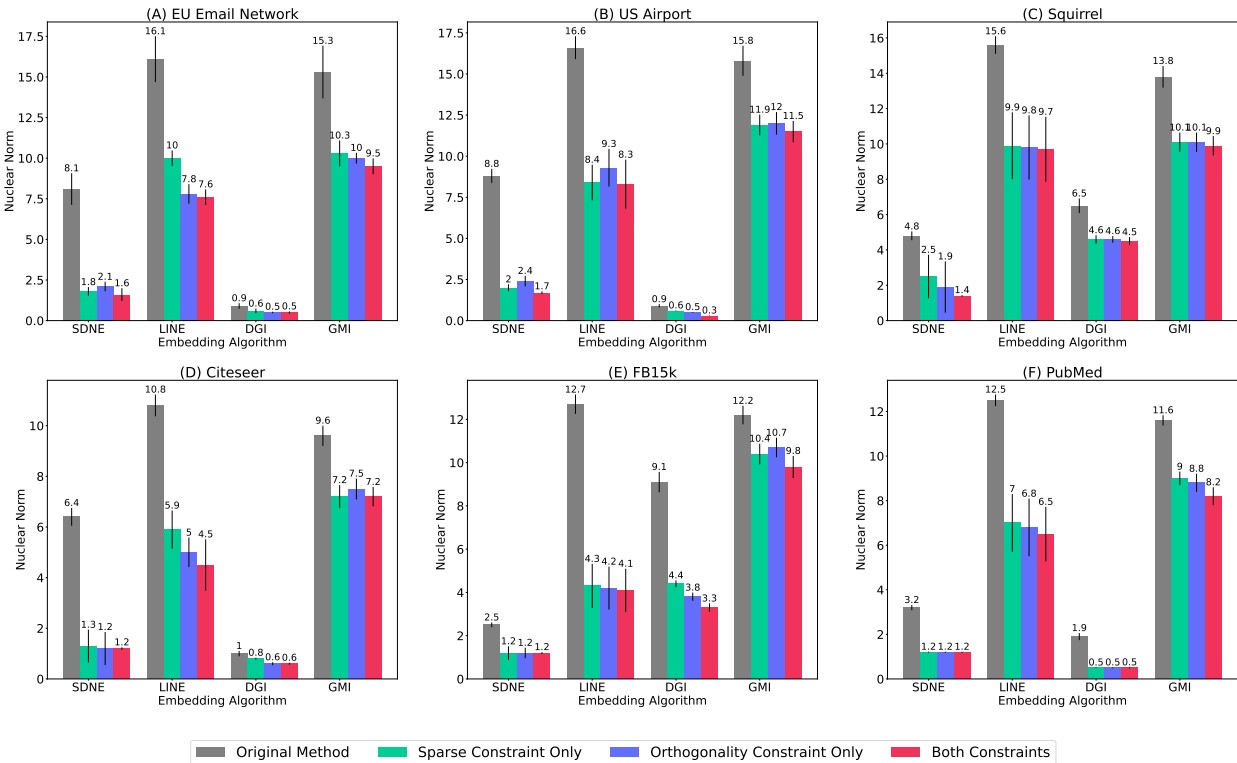

Figure 12: Ablation study examining the impact of each term in the loss function. Observe that the orthogonality constraint alone reduces nuclear norm more than the sparsity constraint alone. However, using both constraints leads to the lowest nuclear norms. The plot shows average values over 5 runs. Results shown are for 128 dimensional embeddings but hold true across various dimensions. (We tried $d = 16, 32, 64, 128, 256$.) We do not find a correlation between the number of embedding dimensions and the reduction in the nuclear norm values. Error bars show standard error.

# 6 Conclusion and Future Work

We present a method for generating explanations for each node in the form of an Explain matrix using a set of human-understandable "sense" features. These features can be based on the graph structure (such as degree, clustering coefficient, and PageRank) or other user-defined features. We use the nuclear norm of the generated Explain matrices to quantify their usefulness. The choice of nuclear norm is due to its relationship with the entropy of a matrix. In particular, reducing the nuclear norm leads to the reduction of the lower bound of the entropy of the Explain matrices. Moreover, we introduce the XM framework, which modifies existing embedding algorithms to produce embeddings with low nuclear norms on Explain matrices, while maintaining downstream task performance. We demonstrate the effectiveness of XM across four embedding algorithms and six real-world datasets and analyze the impact of XM's constraints through an ablation study. XM is independent of any downstream task and can be used with any existing embedding algorithm and any set of sense features.

**Future work.** We noticed that the Explain matrices tend to have dimensions that do not contribute to explaining any sense features – i.e. the rows corresponding to these dimensions in the Explain matrix are near zero. Such information can be used for model selection – i.e. to set the hyperparameter associated with the number of embedding dimensions.

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

## A  Appendix

### A.1  Variations to the Objective Function

Recall that we define the Explain matrix for a node $k$:

$$E_k = \frac{\vec{y_k} \otimes \vec{f_k}^T}{\|\vec{y_k}\|\|\vec{f_k}\|} \tag{21}$$

Thus, the sparsity and orthogonality constraints (as defined in Equations (2) and (3), respectively) are on nodes. To add pairwise constraints on nodes, we investigate variations to Equation (3).

Specifically, we implement the following variations for any node pair $i$ and $j$ :

1. Minimize $a_{ij}\|E_i - E_j\|_F$.
   This constrains the $E$ matrices of each pair of nodes $i, j$ to be similar if they are connected by an edge (i.e. if $a_{ij} = 1$), where the similarity is captured in terms of the Frobenius norm of the difference between their two Explain matrices.

2. Minimize $\|E_i E_j^T - f_i f_j^T\|_F$.
   Here, we move away from asking whether or not two nodes are connected by an edge. We compute the dimension-wise similarities between the Explain matrices and the similarities between sense feature vectors for all possible node pairs and obtain two distance matrices, each of dimension $\mathbb{R}^{n \times n}$. We then minimize the Frobenius norm of the difference between these two matrices.

We find that adding pairwise constraints to the objective function is not a good idea. The outcome is very sensitive to the choice of hyperparameters and the reduction in nuclear norm is not as significant. We posit that this may be due to opposing components of the objective function. For example, consider a high-degree node (a.k.a. a hub) that is connected to a low-degree node. In this case, most algorithms place the embeddings of the two nodes close to each other (since the nodes are connected) even though their sense features are different. To summarize, we find that using the formulation defined in Equations (2) and (3) leads to the most stable setup with the largest reduction in nuclear norms among the ones described above.

## A.2   Nuclear Norm Distributions for Explain Matrices

Recall that each node has a corresponding Explain matrix. Thus, for each algorithm and dataset, we calculate the nuclear norm distribution across the Explain matrices of nodes. Figures 13, 14, and 15, respectively, show the nuclear norm distributions between LINE and LINE+XM, DGI and DGI+XM, and GMI and GMI+XM for all 6 datasets. We observe that the XM variants produce Explain matrices whose nuclear norms are lower than the original versions.

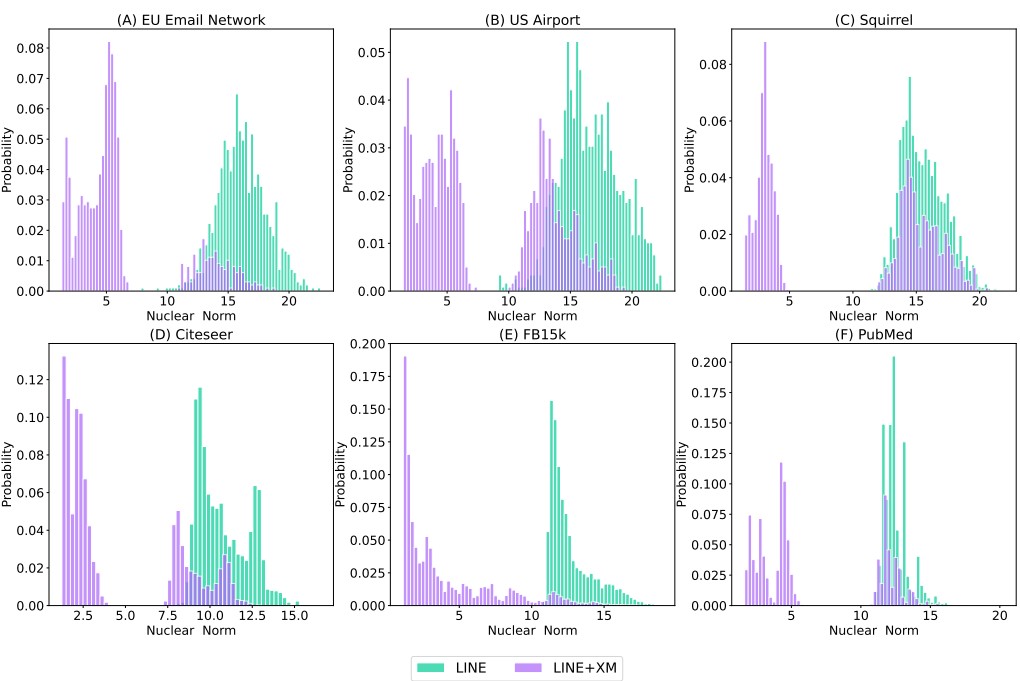

Figure 13: Nuclear norm distribution across the Explain matrices for LINE and LINE+XM. Observe how the distribution of LINE+XM is shifted to the left with a lower mean across each of the 6 datasets.

## A.3   Runtime Analysis

We examine differences in runtime per epoch (in seconds) between the original algorithms and the XM variants across different embedding dimensions. Figure 16 shows the runtime comparison for the US Airport network, Figure 17 for the Squirrel network, Figure 18 for the Citeseer network, Figure 19 for the FB15k network, and Figure 20 for the PubMed network. We observe that the XM variants are comparable in runtime to their original versions.

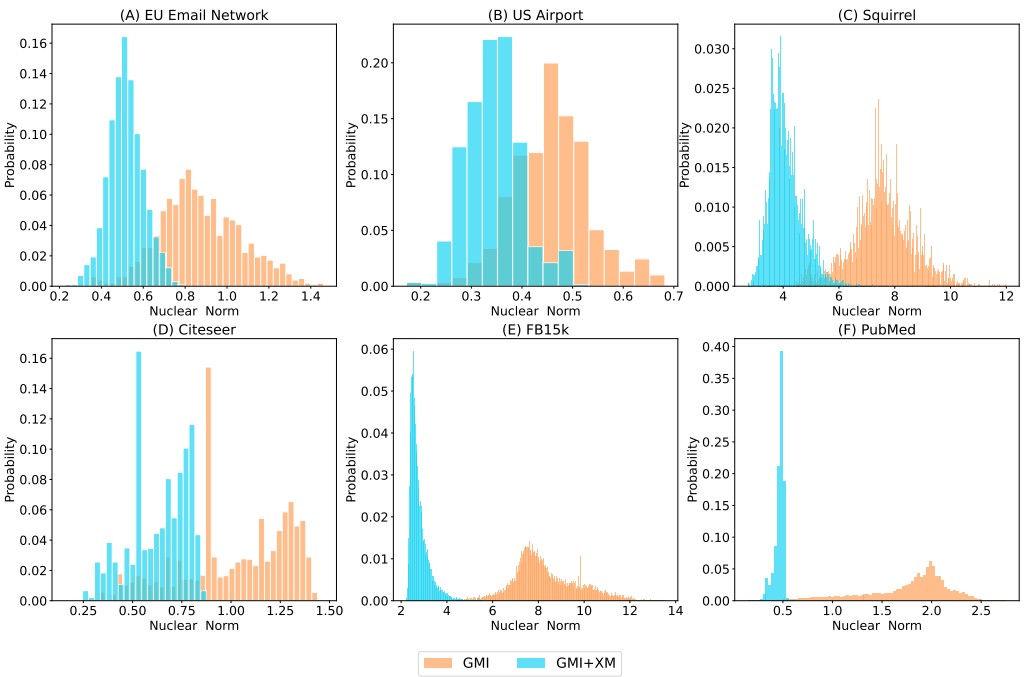

Figure 14: Nuclear norm distribution across the Explain matrices for DGI and DGI+XM. Observe how the distribution of DGI+XM is shifted to the left with a lower mean across each of the 6 datasets.

## A.4   Nuclear Norm Minimization

Figure 21a looks at all networks in Table 3 embedded using each of the 4 algorithms from Table 2. Results are averaged across 5 runs. We see that the nuclear norms of the Explain matrices are lower for the XM variants across each network. These differences are statistically significant with $p$-values less than 0.05. Table 21b lists the $p$-values.

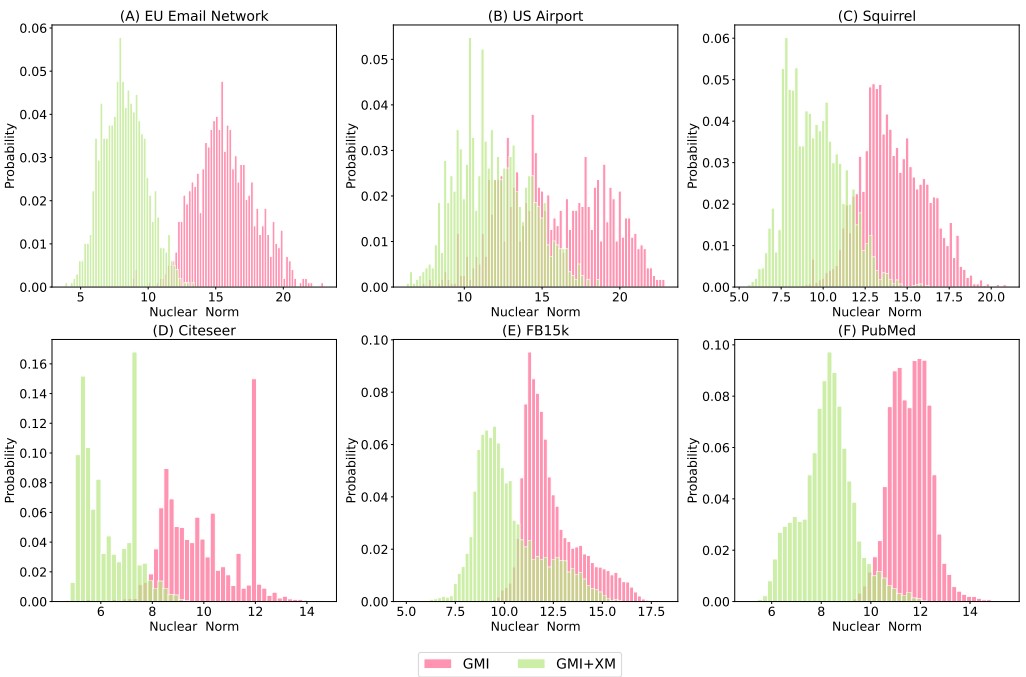

Figure 15: Nuclear norm distribution across the Explain matrices for GMI and GMI+XM. Observe how the distribution of GMI+XM is shifted to the left with a lower mean across each of the 6 datasets.

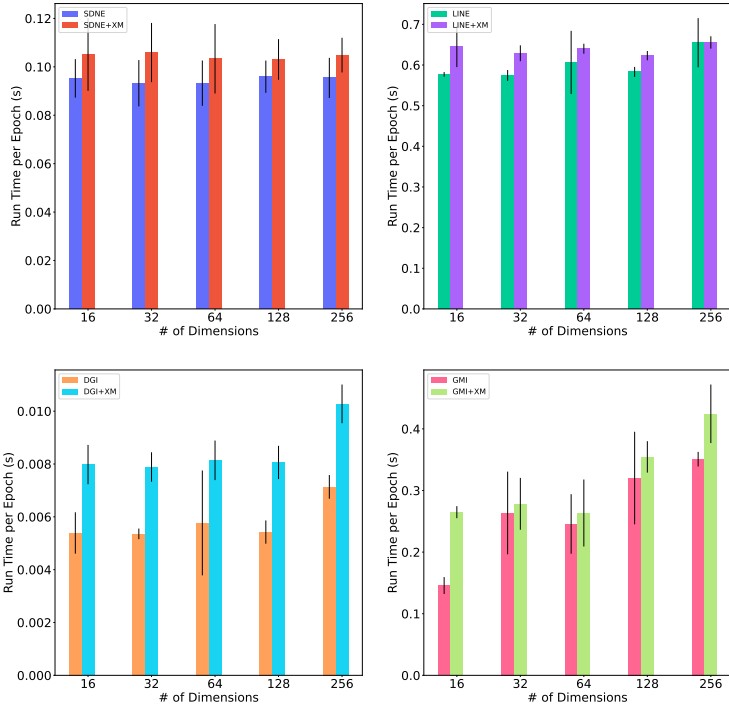

Figure 16: US Airport network: runtimes per epoch (in seconds) for SDNE, LINE, DGI, GMI, and their corresponding XM variants. Results are averaged across 5 runs. Observe that the runtimes per epoch are comparable to the original versions. Error bars show standard error.

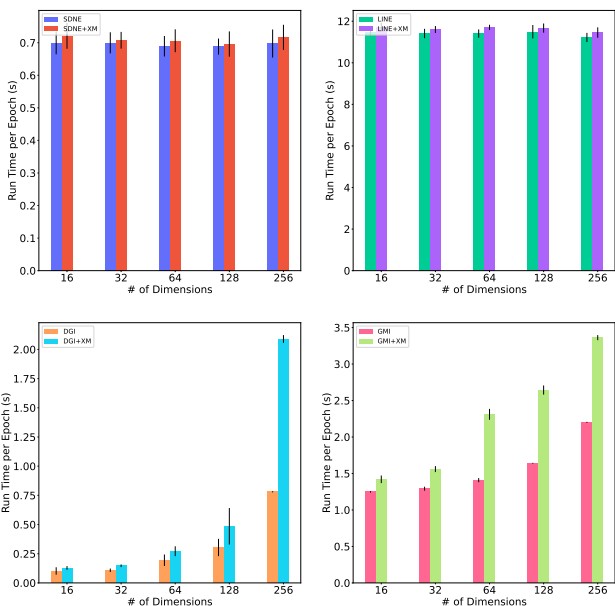

Figure 17: Squirrel network: runtimes per epoch (in seconds) for SDNE, LINE, DGI, GMI, and their corresponding XM variants. Results are averaged across 5 runs. Observe that the runtimes per epoch are comparable to the original versions. Error bars show standard error.

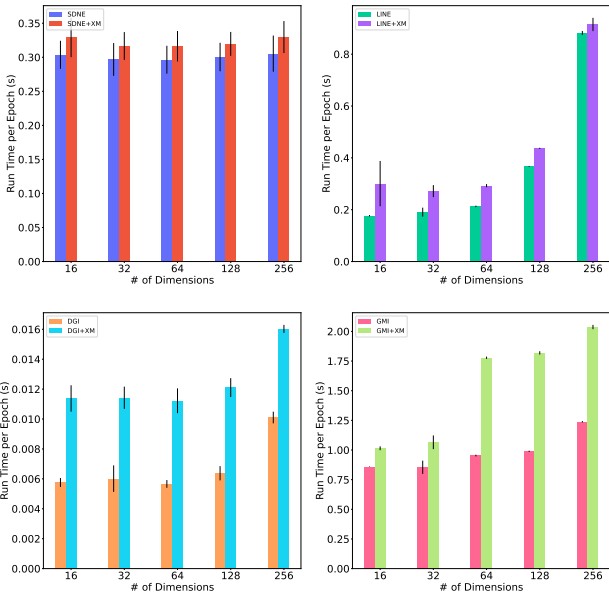

Figure 18: Citeseer network: runtimes per epoch (in seconds) for SDNE, LINE, DGI, GMI, and their corresponding XM variants. Results are averaged across 5 runs. Observe that the runtimes per epoch are comparable to the original versions. Error bars show standard error.

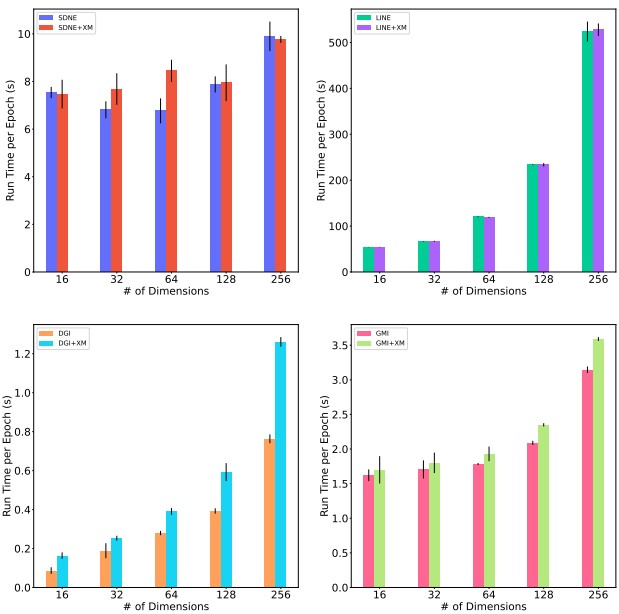

Figure 19: FB15k network: runtimes per epoch (in seconds) for SDNE, LINE, DGI, GMI, and their corresponding XM variants. Results are averaged across 5 runs. Observe that the runtimes per epoch are comparable to the original versions. Error bars show standard error.

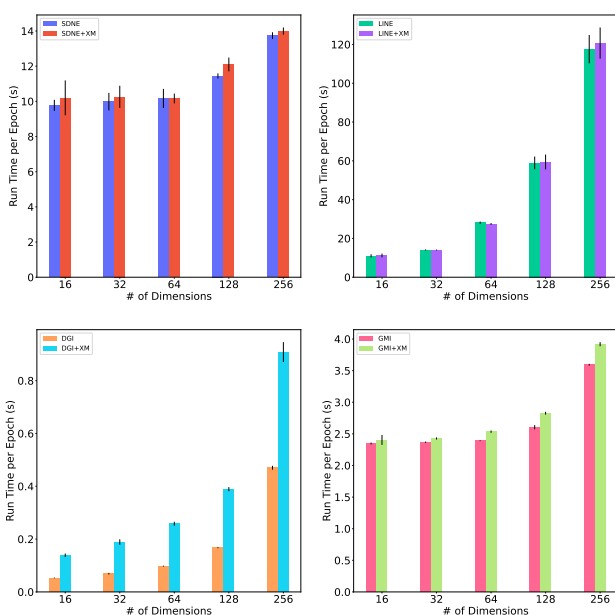

Figure 20: PubMed network: runtimes per epoch (in seconds) for SDNE, LINE, DGI, GMI, and their corresponding XM variants. Results are averaged across 5 runs. Observe that the runtimes per epoch are comparable to the original versions. Error bars show standard error.

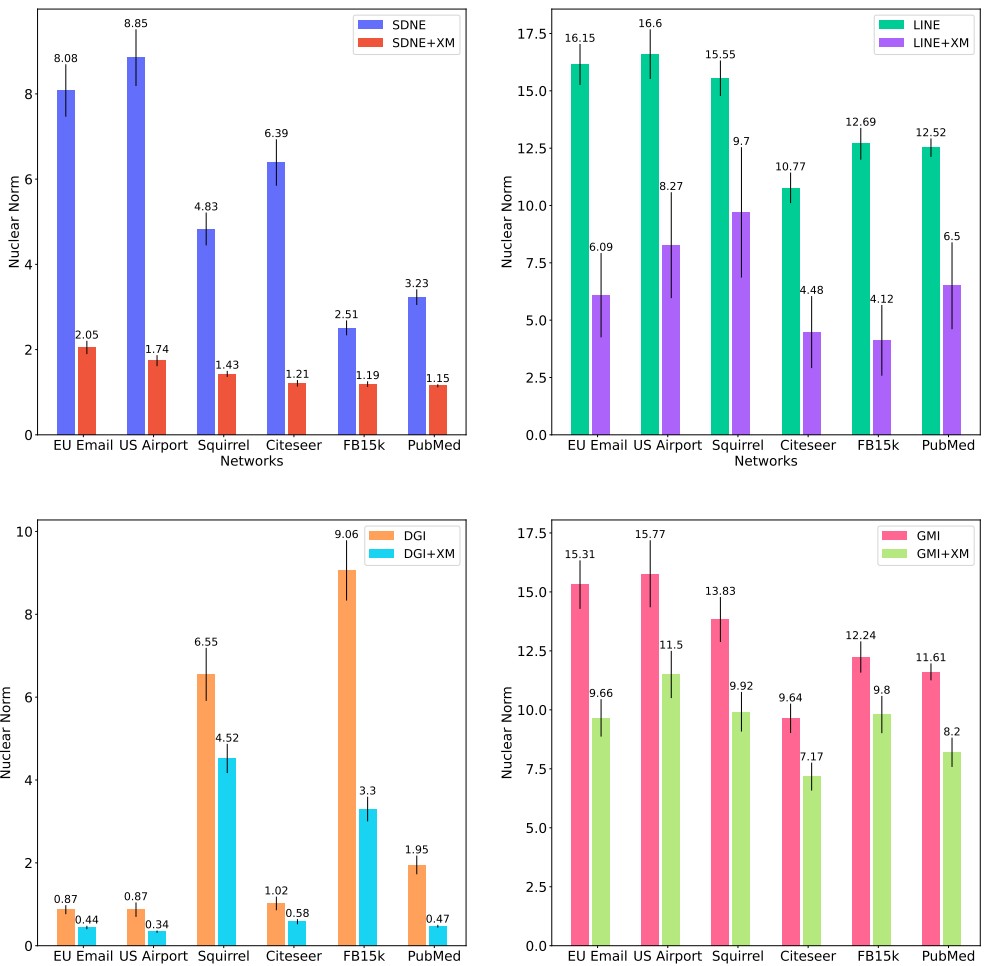

(a) We average the nuclear norms of the Explain matrices of each node in the network. The experiment is repeated 5 times and average results are shown. Error bars depict standard error. Observe that the mean nuclear norms for the XM variants are lower than the original versions. The differences in means are statistically significant with $p$-values less than 0.05.

|  | EU Email | US Airport | Squirrel | Citeseer | FB15k | PubMed |
|---|---|---|---|---|---|---|
| SDNE vs. SDNE + XM | 2.13e-10 | 9.78e-11 | 1.65e-10 | 7.76e-10 | 4.44e-11 | 3.96e-11 |
| LINE vs. LINE + XM | 1.96e-07 | 8.86e-06 | 6.72e-05 | 8.30e-06 | 1.38e-07 | 3.92e-06 |
| DGI vs. DGI + XM | 1.62e-03 | 1.47e-04 | 2.62e-04 | 4.88e-04 | 1.93e-07 | 5.03e-07 |
| GMI vs. GMI + XM | 1.80e-06 | 6.59e-06 | 3.81e-06 | 1.02e-07 | 2.88e-05 | 1.60e-07 |

(b) $p$-values associated with Figure 21a. The nuclear norms between the original versions and the XM variants are statistically significant.

Figure 21: Quantitative evaluation of the Explain matrices based on their nuclear norms across networks, original methods and their XM variants. The lower the norm, the better the Explain matrix.

