# OpenReview forum: "Explaining Node Embeddings"
_TMLR — Accepted by TMLR_

### Review · Reviewer_ZtPJ · 2024-08-05

**Summary Of Contributions:**

The authors introduce a framework for producing explainable node embeddings by minimizing the nuclear norm.  This approach can be shown to reduce the nuclear norm for a number of embedding approaches with comparable computational performance to other methods.  This approach is said to produce more explainable node embeddings, however, this is not well explored or demonstrated in the paper.

**Audience:**

No

**Claims And Evidence:**

No

**Requested Changes:**

Allowing the text to wrap on the x-axis of Figure 1 would make it more readable.

The barbell and karate club examples may be too much of toy examples- I would suggest picking one of them, and one actual real world example- the Karate Club Network is a fine dataset, but it's really not a "real world" dataset.  Actual real world graph problems have hundreds or thousands of nodes.  I would suggest keeping the karate club as the toy example and dropping the dumbbell example as I don't know that it is that illuminating.

The discussion in "Structural vs. Positional Sense Features" is hard to follow.  In part this is because of the formatting which puts the Figure on the next page.  However, the text could also be improved.  The sentence such as "We define a second set of positional features" seems out of nowhere- where was the first set of defined (do we assume it's the full subset up in the first paragraph of 2.1)? What are they for this example?

The subset of chosen sense features was selected because of the correlation among the broader set.  However, from Figures 1 and 2, the remaining features still appear to be highly redundant.

I would consider leading with some kind of tangible example in the introduction.  Perhaps pick a network, show what the embeddings look like by current approaches, show what can be understood given the approach used in this method.  In its current formulation, interpretability is the key advantage, but what can be interpreted and what value that offers does not come until the very end of the paper.

3.2 needs further explanation.  It's shown that the distribution shifts and narrows, but what does this tell the reader?  Essentially this result needs to be interpreted.  Similarly 3.3 needs better explanation.  It is stated that the nuclear norms are lower- explain the "so what" to the reader.

The opening of the discussion is entirely unclear- "the explanations are granular".  I don't understand what this means.

While it can go anywhere, I think putting the related works after the intro might help the narrative in this case.

**Strengths And Weaknesses:**

STRENGHTS:
Model interpretability is a valuable area of study.  This approach shows general reduction in the nuclear norm and higher AUC with minimal computation time for several different embedding algorithms.

Trials are run multiple times to produce errorbars.


WEAKNESSES:
The text needs to be cleaned up to flow better and remove redundancy, particularly in the introduction.  Each sentence is largely well structured, but the overall arc to the content could be improved.  Other sections of the next need improvements to clarity and explanation as well (see Requested Changes).  In general, the text needs to do a better job bringing the reader along on the journey- we did X, we saw Y result, this demonstrates Z.

The benefit of this approach is supposed to be around interpretability, but this is never explored- the results are never interpreted.  It's unclear after reading the paper what the user of this approach can do that they couldn't do without it.

It's mentioned that the choice of features is very important, but this is not explored.  There needs to be a set of experiments which shows how the choice of features impacts the results.

Comparison to other methods is not explored.  All comparisons are shown in terms of Embeddings w/wo this approach.  Some of the other fundamentally different approaches such as those covered in the related works should be shown.

---

> ### Author Response · Authors · 2025-01-27
>
> Thank you for the constructive review. We revised the manuscript to address the review. Our revisions are highlighted in blue. Here is a summary:
> 1. We updated the manuscript to address all the readability and presentation concerns.
> 2. We updated the Introduction Section with additional motivation and problem definition.
> 3. We moved and rewrote the Related Works Section.
> 4. We updated Section 3.1.2 to better motivate the choice of sense features.
> 5. We updated Sections 3.2.2 and 3.2.4 on the use of nuclear norm.
> 6. We added a case study in Section 4.4 to highlight how XM can be used in real world scenarios and compare against 2 additional methods.
> 7. We claim to answer 2 questions in this manuscript (Q1 and Q2 in the Introduction) and provide evidence to that end.
> 8. We respectfully disagree that our work is not suitable for TMLR audience. TMLR has published many papers on interpretability of ML models. The last one (that we know of) was in October 2024 (https://openreview.net/pdf?id=yHUtuvoIQv).

---

### Review · Reviewer_czVb · 2024-08-08

**Summary Of Contributions:**

This paper presents a method for understanding what each dimension of a node embedding method means, with respect to a set of human-understandable sense features defined on nodes. To evaluate Explain matrices, their nuclear norms are minimized.

**Audience:**

Yes

**Claims And Evidence:**

Yes

**Requested Changes:**

Please see the comments above for details.

**Strengths And Weaknesses:**

This is an interesting are of research study that could be highly useful for future research in the field and model development since human interpretability is one component lost in the learning process for machine learning models.

That said, the paper needs to be organized in a more effective manner. If the Related Work section is moved after the Introduction, the clarity of the work improves. Moreover, the authors should consider including Tables to summarize the key properties that their model contributes, and the properties of other baseline models in terms of challenges and what they accomplish. This would help to better motivate the work.

There needs to be a problem formulation section to summarize what the graph looks like namely by what do the nodes/edges represent.
The sense features of this work are interesting, but it just seems like engineered features which one can directly compute based on the structure of the graph e.g., by degree, and betweenness centrality.

Furthermore, it is unclear how we know the 15 hand-selected features are the best explainable or observed features for the task, since latent embeddings are more flexible to learn but can also learn hidden properties or correlations in the dataset that do not map to any observed feature.

---

> ### Author Response · Authors · 2025-01-27
>
> Thank you for the constructive review. We revised the manuscript to address the review. Our revisions are highlighted in blue. Here is a summary:
>
> 1. We moved the Related Works Section after the Introduction and added a taxonomy of explanation methods covered in the Related Works Section. We reorganized that section to match the taxonomy.
> 2. We added a formal problem definition and a notation table to the Introduction Section.
> 3. We added a paragraph to the Introduction Section that provides a motivation and the origins of the sense features.
> 4. We edited Section 3.1.2 to better motivate the choice of sense features.
> 5. We do not want a one-to-one correspondence between sense features and embedding dimensions, as this would degrade downstream task utility. We have highlighted this point in Section 3.2.3 and cite relevant literature.
> 6. XM’s hyperparameters can be tuned to get embeddings that are both interpretable and have downstream task utility. This tradeoff is highlighted in Figure 7.

---

### Review · Reviewer_5ybi · 2025-01-13

**Summary Of Contributions:**

# Contribution
Created a framework called XM (Explain Embedding) to improve existing node embeddings (can be any 3P embedding methods: using DGI in paper) to be as close as possible to human-understandable embeddings. The XM framework is not purely a post-hoc method for human-understandable embeddings but also manipulates the generated embeddings to incorporate interpretability during the embedding process.

## **XM as post-hoc**

- **Input**
   -  Graph Embeddings using any 3rd party embedding methods such as DGI

- **Sense Features**
   - Define a set of “sense” features which are human-understandable that describes the graph: {degree, clustering coefficient, Katz centrality.}

- **Correlation of Embeddings with sense features**
  - Compare the precomputed embeddings of graph nodes with these sense features to understand how each embedding dimension correlates with specific graph properties

- **Explain Matirx (E)**
   - Outputs an Explain Matrix (E) that shows how each embedding dimension aligns with each sense feature, making the embedding interpretable: how graph nodes are connected based on these sense features.

## **XM as Embedding Manipulator**

 XM as an Embedding Manipulator (with sense features) to generate embeddings for Nuclear Norm Minimization  (to keep the
 embeddings as human-understandable) to keep the simpler relationship between the sense features and embedding but at the same
  time avoiding a linear connection:

- **Select Embedding Methods**
    - for demonstration, the author is using  4 unsupervised embedding algorithms: 2 with node attributes and 2 without node attributes

- **Define Sense Features**

- **Augment EMbeddings**
    - Augment original embedding with sense features with two constraints to reach nuclear norm minimization
    - Sparsity: each embedding dimension focuses on the subset of features to remove noise
    - Orthogonality:  Ensure embedding dimensions are diverse

- **Reduce Nuclear Norm**
   - Train the embeddings to reduce the nuclear norm to explain Metrix.

The paper presents a good amount of examples/case studies on XM.

**Audience:**

Yes

**Broader Impact Concerns:**

- **Bias In Sense Feature selection**
   - The XM frameworks rely on the selection of sense features, poorly chosen sense features will lead to irrelevant explanations.
   - Recommend having "Best Practices" on choosing these sense features and how to adopt these features when the usecase is different for example: multimodality (text, videos, images)

- **Interoperability vs accountability**
     - XM depends heavily on sense features and that might improve the interoperability but fail to guarantee accountability. Example: A biased sense feature may incorrectly highlight certain groups or nodes as "less central" or "less connected," influencing decisions in hiring, credit scoring, or law enforcement.

**Claims And Evidence:**

Yes

**Requested Changes:**

- Conduct some experiments on how the selection of sense features   might impact the interpretability
- To diversify the approach, provide some guidelines on how these sense features are directly connected to the use case and depending on the use case how a user should select these features.
-  If there are any examples that showcase the sensitivity of the nuclear norm with different hyperparameter choices: are there any scenarios where focusing on reducing the nuclear norm might also be removing the important embeddings

**Strengths And Weaknesses:**

## **Strengths**

-  The XM algorithm introduces a novel use of sparsity and orthogonality constraints to align graph embeddings with human-understandable sense features, providing a significant contribution to the field of interpretable graph representation learning.

## **Weakness**

- **Fixed sense features**
     - The XM framework chooses a fixed type of sense features and with that the explanation makes sense. The accuracy of XM is highly dependent on these sense features and as per the author, poor choice of sense features might degrade the explanation. For users, it could have been helpful on the best practices to select these predefined sense features along with defining “new” type of features depending on the use-case for example, what can help with multi-modality.

- **Nuclear Norm Reduction is the "only" Interpretability Method**
   - The author is completely relying on nuclear norm reduction for interpretability, but the nuclear norm reduction doesn’t ensure the “quality” or meaningfulness of each embedding. So complementing the nuclear norms with other matrices (defining the usefulness of embedding for each use-case) might provide a more robust solution.

---

> ### Author Response · Authors · 2025-01-27
>
> Thank you for the constructive review. We revised the manuscript to address the review. Our revisions are highlighted in blue. Here is a summary:
> 1. We edited Section 3.1.2 to better motivate the choice of sense features.
> 2. We chose structural sense features because these features are well-established in domains that use graph data (e.g., network science, social network analysis, and graph mining).
> 3. Focusing only on nuclear norm reduction would indeed lead to a reduction on downstream task performance. We do not want a one-to-one correspondence between sense features and embedding dimensions, as this would degrade downstream task utility. We have highlighted this point in Section 3.2.3 and cite relevant literature.
> 4. XM’s hyperparameters can be tuned to get embeddings that are both interpretable and have downstream task utility. This tradeoff is highlighted in Figure 7.
> 5. Our code is publicly available, detailing hyperparameters used for each dataset and algorithm combination.

---

### Decision · Action_Editor_f3bL · 2025-03-31

**Recommendation:** Accept as is

**Comment:**

As I mentioned, I am marking the paper as "Accept" but I would encourage authors to use the opportunity to polish the paper a bit more before uploading camera-ready version.

**Audience:**

The paper is highly relevant to TMLR community as both embedding and explanations are primary concerns of ML community. I believe the paper would find a broad audience .

**Claims And Evidence:**

This paper attempts to provide explanations for embedding and consequently modifies the existing algorithms to achieve these embeddings. The ideas in the original submission were found to be interesting but the reviewers felt that the work would need more empirical experiments. The authors have performed these and consequently all the reviewers agree that the paper convincingly supports the claims.

One of the reviewers have urged authors to refine the text further. I think this is a reasonable suggestion and I believe that the authors would use the opportunity to further refine the text: as the typical revisions in the review process seem to add text and perhaps the final version can be an opportunity to remove some of the text for clarity.